# Robust Bayesian Satisficing

**Artun Saday**
Bilkent University
`artun.saday@bilkent.edu.tr`

**Yaşar Cahit Yıldırım**
Bilkent University
`cahit.yildirim@bilkent.edu.tr`

**Cem Tekin**
Bilkent University
`cemtekin@ee.bilkent.edu.tr`

## Abstract

Distributional shifts pose a significant challenge to achieving robustness in contemporary machine learning. To overcome this challenge, robust satisficing (RS) seeks a robust solution to an unspecified distributional shift while achieving a utility above a desired threshold. This paper focuses on the problem of RS in contextual Bayesian optimization when there is a discrepancy between the true and reference distributions of the context. We propose a novel robust Bayesian satisficing algorithm called RoBOS for noisy black-box optimization. Our algorithm guarantees sublinear lenient regret under certain assumptions on the amount of distribution shift. In addition, we define a weaker notion of regret called robust satisficing regret, in which our algorithm achieves a sublinear upper bound independent of the amount of distribution shift. To demonstrate the effectiveness of our method, we apply it to various learning problems and compare it to other approaches, such as distributionally robust optimization.

## 1 Introduction

*Bayesian optimization* (BO) [1, 2] is a powerful technique for optimizing complex black-box functions that are expensive to evaluate. It is particularly useful in situations where the function is noisy or has multiple local optima. The approach combines a probabilistic model of the objective function with a search algorithm to efficiently identify the best input values. In recent years, BO has become a popular method for various sequential decision-making problems such as parameter tuning in machine learning [3], vaccine and drug development [4], and dynamic treatment regimes [5].

Contextual BO [6] is an extension of BO that allows for optimization in the presence of contexts, i.e., exogenous variables associated with the environment that can affect the outcome. A common approach to BO when the context distribution is known is to maximize the expected utility [7, 8]. Often, however, there exists a distributional mismatch between the reference distribution that the learner assumes and the true covariate distribution the environment decides. When there is uncertainty surrounding the reference distribution, choosing the solution that maximizes the expected utility may result in a suboptimal or even disastrous outcome. A plethora of methods have been developed to tackle distribution shifts in BO; the ones that are most closely related to our work are *adversarially robust* BO (STABLEOPT) [9] and *distributionally robust* BO (DRBO) [10]. STABLEOPT aims to maximize the utility under an adversarial perturbation to the input. DRBO aims to maximize the utility under the worst-case context distribution in a known uncertainty set. We focus on the contextual framework of DRBO. However, unlike DRBO, our algorithm does not require as input an uncertainty set. This provides an additional level of robustness to distribution shifts, even when the true distribution lies outside of a known uncertainty set.

37th Conference on Neural Information Processing Systems (NeurIPS 2023).

In this paper, we introduce the concept of *robust Bayesian satisficing* (RBS), whose roots have been set by Herbert Simon [11]. In his Nobel Prize in Economics speech in 1978, Simon mentioned that "decision makers can satisfice either by finding optimum solutions for a simplified world or by finding satisfactory solutions for a more realistic world". Satisficing can be described as achieving a satisfactory threshold $\tau$ (aka aspiration level) utility under uncertainty. It has been observed that satisficing behavior is prevalent in decision-making scenarios where the agents face risks and uncertainty and exhibit bounded rational behavior due to the immense complexity of the problem, computational limits, and time constraints [12]. Since its introduction, satisficing in decision-making has been investigated in many different disciplines, including economics [13], management science [14, 15], psychology [16], and engineering [17, 18]. The concept of satisficing has also been recently formalized within the multi-armed bandit framework in terms of regret minimization [19, 20, 21, 22] and good arm identification [23]. Recently, it has been shown that satisficing designs can be found with a sample complexity much smaller than what is necessary to identify optimal designs [23]. Moreover, [21] demonstrated that when the future rewards are discounted, i.e., learning is time-sensitive, algorithms that seek a satisficing design yield considerably larger returns than algorithms that converge to an optimal design. The concept of *robust satisficing* (RS) is intimately connected to satisficing. In the seminal work of Schwartz et al. [24], robust satisficing is described as finding a design that maximizes the robustness to uncertainty and satisfices. Long et al. [25] cast robust satisficing as an optimization problem and propose models to estimate a robust satisficing decision efficiently.

Inspired by this rich line of literature, we introduce the concept of *robust Bayesian satisficing* (RBS) as an alternative paradigm for robust Bayesian optimization. The objective of RBS is to satisfice under ever-evolving conditions by achieving rewards that are comparable with an aspiration level $\tau$, even under an unrestricted distributional shift.

**Contributions.** Our contributions can be summarized as follows:

- We propose *robust Bayesian satisficing*, a new decision-making framework that merges *robust satisficing* with the power of Bayesian surrogate modeling. We provide a detailed comparison between RBS and other BO methods.

- To measure the performance of the learner with respect to an aspiration level $\tau$, we introduce two regret definitions. The first one is the *lenient regret* studied by [22] and [23]. The second one is the *robust satisficing regret*, which measures the loss of the learner with respect to a benchmark that tracks the quality of the true robust satisficing actions. We also provide a connection between these two regret measures.

- We propose a Gaussian process (GP) based learning algorithm called *Robust Bayesian Optimistic Satisficing* (RoBOS). RoBOS only requires as input an aspiration level $\tau$ that it seeks to achieve. Unlike algorithms for robust BO, it does not require as input an uncertainty set that quantifies the degree of distribution shift.

- We prove that RoBOS achieves with high probability $\tilde{O}(\gamma_T \sqrt{T})$ robust satisficing regret and $\tilde{O}(\gamma_T \sqrt{T} + E_T)$ lenient regret, where $\gamma_T$ is the maximum information gain over $T$ rounds and $E_T := \sum_{t=1}^{T} \epsilon_t$ is the sum of distribution shifts by round $T$, where $\epsilon_t$ is the amount of distribution shift in round $t$.

- We provide a detailed numerical comparison between RoBOS and other robust BO algorithms, verifying the practical resilience of RoBOS in handling unknown distribution shifts.

**Organization.** The remainder of the paper is organized as follows. Problem formulation and regret definitions are introduced in Section 2. RoBOS is introduced in Section 3, and its regret analysis is carried out in Section 4. Experimental results are reported in Section 5. Conclusion, limitations, and future research are discussed in Section 6. Additional results and complete proofs of the theoretical results in the main paper can be found in the appendix.

## 2   Problem definition

Let $f : \mathcal{X} \times \mathcal{C} \to \mathbb{R}$ be an *unknown* reward function defined over a parameter space with finite action and context sets, $\mathcal{X}$ and $\mathcal{C} := \{c_1, \dots, c_n\}$ respectively. Let $\mathcal{P}_0$ represent the set of all distributions

Table 1: Comparison of optimization objectives.

| Method | Inputs | Objective |
|--------|--------|-----------|
| SO | $f, P_t$ | Find $x \in \mathcal{X}$ that maximize $\mathbb{E}_{c \sim P_t}[f(x,c)]$ |
| S | $f, P_t, \tau$ | Find $x \in \mathcal{X}$ that satisfy $\mathbb{E}_{c \sim P_t}[f(x,c)] \geq \tau$ |
| WRO | $f, P_t, \Delta_t$ | Find $x \in \mathcal{X}$ that maximize $\min_{c \in \Delta_t} f(x,c)$ |
| DRO | $f, \mathcal{U}_t$ | Find $x \in \mathcal{X}$ that maximize $\inf_{P \in \mathcal{U}_t} \mathbb{E}_{c \sim P}[f(x,c)]$ |
| RS | $f, P_t, \tau$ | Find $x \in \mathcal{X}$ that minimize $k(x)$ where |
|  |  | $k(x) = \min k$ s.t. $\mathbb{E}_{c \sim P}[f(x,c)] \geq \tau - k\Delta(P, P_t),\ \forall P \in \mathcal{P}_0,\ k \geq 0$ |

over $\mathcal{C}$. The objective is to sequentially optimize $f$ using noisy observations. At each round $t \in [T]$, in turn the environment provides a reference distribution $P_t \in \mathcal{P}_0$, the learner chooses an action $x_t \in \mathcal{X}$ and the environment provides a context $c_t \in \mathcal{C}$ together with a noisy observation $y_t = f(x_t, c_t) + \eta_t$, where $\eta_t$ is conditionally $\sigma$-subgaussian given $x_1, c_1, y_1, \ldots, x_{t-1}, c_{t-1}, y_{t-1}, x_t, c_t$. We assume that $c_t$ is sampled independently from a time-dependent, unknown distribution $P_t^* \in \mathcal{P}_0$, which can be different than $P_t$. We represent distributions $P_t$ and $P_t^*$ with $n$-dimensional non-negative vectors $w_t$ and $w_t^*$ such that $||w_t||_1 = ||w_t^*||_1 = 1$. We represent the distance between $P, P' \in \mathcal{P}_0$ with $\Delta(P, P')$. In particular, we consider *maximum mean discrepancy* (MMD) as the distance measure.

**Optimization objective.** To motivate robust Bayesian satisficing, we review and compare various optimization objectives. Throughout this section, we assume that $f$ is known. The key novelty of our work is combining the new *robust satisficing* objective from [25] with Bayesian surrogate modeling to address unmet real-world challenges faced by BO. *Robust satisficing* aims to perform satisfactorily well over a wide range of possible distributions on $\mathcal{C}$. This is different from *stochastic optimization* (SO) [26] which aims to optimize for a given reference distribution $P_t$,[1] *distributionally robust optimization* (DRO) [27, 28], which aims to optimize the worst-case scenario in an ambiguity set $\mathcal{U}_t$, usually taken as a ball of radius $r$ centered at $P_t$, and *worst-case robust optimization* (WRO) [29], which aims to optimize under the worst-case contexts from an uncertainty set $\Delta_t$ of contexts, and *satisficing* (S) [30] which seeks for a satisfactory solution that achieves threshold $\tau$. Table 1 compares different optimization objectives. In RS, the objective is to find $x_t^* \in \mathcal{X}$ that solves in each round $t$

$$\kappa_{\tau,t} = \min k \text{ s.t. } \mathbb{E}_{c \sim P}[f(x,c)] \geq \tau - k\Delta(P, P_t),\ \forall P \in \mathcal{P}_0,\ x \in \mathcal{X},\ k \geq 0. \tag{1}$$

To find $x_t^*$, we can first compute the *fragility* of $x \in \mathcal{X}$ as

$$\kappa_{\tau,t}(x) = \min k \text{ s.t. } \mathbb{E}_{c \sim P}[f(x,c)] \geq \tau - k\Delta(P, P_t),\ \forall P \in \mathcal{P}_0,\ k \geq 0. \tag{2}$$

When (1) is feasible, the RS solution at round $t$ is the one with the minimum fragility, i.e., $x_t^* \in \arg\min_{x \in \mathcal{X}} \kappa_{\tau,t}(x)$ and $\kappa_{\tau,t} = \min_{x \in \mathcal{X}} \kappa_{\tau,t}(x)$. Similar to DRO, RS utilizes a reference distribution assumed to be a proxy for the true distribution. However, unlike DRO, we do not define an ambiguity set $\mathcal{U}_t$ that represents all plausible distributions on $\mathcal{C}$ but rather define a threshold value $\tau$, which we aim to satisfy. In cases where we are not confident that the reference distribution $P_t$ accurately represents the true distribution $P_t^*$, finding a meaningful ambiguity set can be difficult. In contrast, $\tau$ has a meaningful interpretation and can be expressed as a percentage of the SO solution computed under the reference distribution $P_t$ over $\mathcal{C}$ given by[2]

$$Z_t := \max_{x \in \mathcal{X}} \mathbb{E}_{c \sim P_t}[f(x,c)]. \tag{3}$$

Unlike DRO, in RS, we are certain that our formulation covers the true distribution $P_t^*$. The fragility can be viewed as the minimum rate of suboptimality one can obtain with respect to the threshold per unit of distribution shift from $P_t$. The success of an action is measured by whether it achieves the desired threshold value $\tau$ in expectation. Depending on the objective function and the ambiguity set, the RS solution can differ from the DRO and the SO solutions (see Figure 1 for an example).

---

[1] One particular instance of this is when $P_t$ is the empirical context distribution.

[2] When $f$ is unknown, $Z_t$ cannot be computed exactly. However, one can reflect the uncertainty about $f$ by using a GP surrogate model, by which upper and lower bounds on $Z_t$ can be computed.

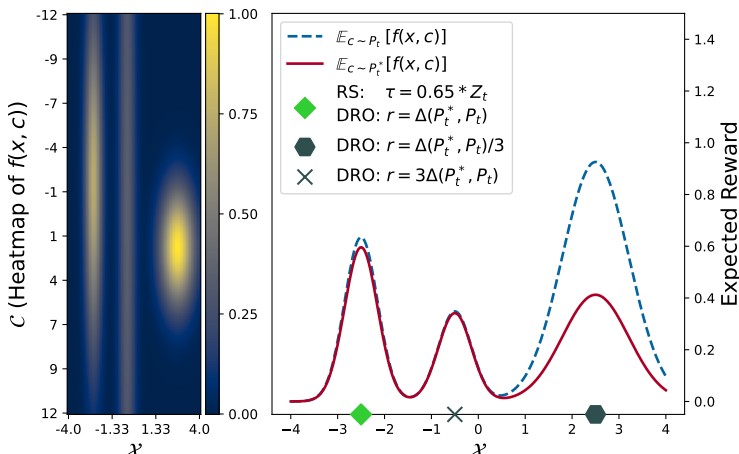

Figure 1: Examples of RS, DRO, and SO solutions where $Z_t$ is the solution to the SO problem given in (3). For DRO, $\mathcal{U}_t$ corresponds to a ball centered at $P_t$ with radius $r$. Rhombus, cross, and hexagon correspond to RS and two other suboptimal solutions, respectively. Note that the SO solution corresponds to the point with a hexagon, i.e., a suboptimal solution. When the radius of the ambiguity ball of DRO captures the discrepancy between $P_t$ and $P_t^*$ perfectly, it selects the RS solution. When the ambiguity ball is too small or too large, it fails to find the RS solution and selects a suboptimal solution.

An intriguing question is whether $\tau$ of RS is more interpretable than $\mathcal{U}_t$ of DRO. While [25] provides a detailed discussion of the pros and cons of these two approaches, below we compare them on an important dynamic drug dosage problem.

**Example 1.** *Type 1 Diabetes Mellitus (T1DM) patients require bolus insulin doses (id) after meals for postprandial blood glucose (pbg) regulation. One of the most important factors that affect pbg is meal carbohydrate (cho) intake [31]. Let $\mathcal{X}$ and $\mathcal{C}$ represent admissible id and cho values. For $x \in \mathcal{X}$, $c \in \mathcal{C}$, let $g(x, c)$ represent the corresponding (expected) bpg value. Function $g$ depends on the patient's characteristics and can be regarded as unknown. The main goal of pbg regulation is to keep pbg close to a target level $K$ in order to prevent two potentially life-threatening events called hypoglycemia and hyperglycemia. This requires $x_t$ to be chosen judiciously based on current $c_t$. Patients rely on a method called cho counting to calculate $c_t$. Often, this method is prone to errors [32]. The reported cho intake $\zeta_t$ can differ significantly from $c_t$. In order to use DRO, one needs to identify a range of plausible distributions for cho calculation errors, which is hard to calculate and interpret. On the other hand, specifying $\tau$ corresponds to defining an interval of safe pbg values around $K$ that one is content with, which is in line with the standard clinical practice [33]. We provide experiments on this application in Section 5.*

**Regret measures.** We consider RS from a regret minimization perspective, where the objective is to choose a sequence of actions $x_1, \ldots, x_T$ that minimize growth rate of the regret. In particular, we focus on two different regret measures. The first one is the *lenient regret* [22] given as

$$R_T^l := \sum_{t=1}^{T} \left( \tau - \mathbb{E}_{c \sim P_t^*}[f(x_t, c)] \right)^+ , \qquad (4)$$

where $(\cdot)^+ := \max\{0, \cdot\}$. The lenient regret measures the cumulative loss of the learner on the chosen sequence of actions w.r.t. specified threshold $\tau$ that we aim to achieve. If an action achieves $\tau$ in the expectation it accumulates no regret, otherwise, it accumulates the difference. Achieving sublinear lenient regret under any distribution shift is an impossible task. Note that even the RS action $x_t^*$ computed with complete knowledge of $f$ only guarantees an expected reward at least as large as $\tau - \kappa_{\tau,t} \Delta(P_t^*, P_t)$. Therefore, our lenient regret upper bound will depend on distribution shifts.

We also define a new notion of regret called *robust satisfying regret*, given as

$$R_T^{rs} := \sum_{t=1}^{T} \left( \tau - \kappa_{\tau,t}\Delta(P_t^*, P_t) - \mathbb{E}_{c \sim P_t^*}[f(x_t, c)] \right)^+ . \tag{5}$$

$R_T^{rs}$ measures the accumulated loss of the learner with respect to the robust satisfying benchmark $\tau - \kappa_{\tau,t}\Delta(P_t^*, P_t)$ under the true distribution. In particular, the true robust satisfying action $x_t^*$ achieves

$$\mathbb{E}_{c \sim P_t^*}[f(x_t^*, c)] \geq \tau - \kappa_{\tau,t}\Delta(P_t^*, P_t) .$$

It is obvious that $R_T^{rs} \leq R_T^l$. When there is no distribution shift, i.e., $P_t^* = P_t$ and (1) is feasible for all $t$, then the two regret notions are equivalent.

In order to minimize the regrets in (4) and (5), we will develop an algorithm that utilizes *Gaussian processes* (GPs) as a surrogate model. This requires us to impose mild assumptions on $f$, which are detailed below.

**Regularity assumptions.** We assume that $f$ belongs to a *reproducing kernel Hilbert space* (RKHS) with kernel function $k$. We assume that $k((x, c), (x', c')) = k_{\mathcal{X}}(x, x')k_{\mathcal{C}}(c, c')$ is the product kernel formed by kernel functions $k_{\mathcal{X}}$ and $k_{\mathcal{C}}$ defined over the action and context spaces respectively. Let the Hilbert norm $||f||_{\mathcal{H}}$ of $f$ be bounded above by $B$. This is a common assumption made in BO literature which allows working with GPs as a surrogate model.

**GP surrogate and confidence intervals.** Define $\mathcal{Z} := \mathcal{X} \times \mathcal{C}$. Our algorithm uses a GP to model $f$, defined by a prior mean function $\mu(z) = \mathbb{E}[f(z)]$ and a positive definite kernel function $k(z, z') = \mathbb{E}[(f(z) - \mu(z))(f(z') - \mu(z'))]$. Furthermore we assume $\mu(z) = 0$ and $k(z, z) \leq 1$, $z \in \mathcal{Z}$. The prior distribution over $f$ is modeled as $GP(0, k(z, z'))$. Using Gaussian likelihood with variance $\lambda > 0$, the posterior distribution of $f$, given the observations $\boldsymbol{y}_t = [y_1, \ldots, y_t]^\mathsf{T}$ at points $\boldsymbol{z}_t = [z_1, \ldots, z_t]^\mathsf{T}$ is modeled as a GP with posterior mean and covariance at the beginning of round $t \geq 1$ given as

$$\mu_t(z) = \boldsymbol{k}_{t-1}(z)^\mathsf{T}(\boldsymbol{K}_{t-1} + \lambda \boldsymbol{I}_{t-1})^{-1}\boldsymbol{y}_{t-1}$$
$$k_t(z, z') = k(z, z') - \boldsymbol{k}_{t-1}(z)^\mathsf{T}(\boldsymbol{K}_{t-1} + \lambda \boldsymbol{I}_{t-1})^{-1}\boldsymbol{k}_{t-1}(z')$$
$$\sigma_t^2(z) = k_t(z, z) ,$$

where $\boldsymbol{k}_t(z) = [k(z, z_1) \ldots k(z, z_t)]^\mathsf{T}$, $\boldsymbol{K}_t$ is the $t \times t$ kernel matrix of the observations with $(\boldsymbol{K}_t)_{ij} = k(z_i, z_j)$ and $\boldsymbol{I}_t$ is the $t \times t$ identity matrix. Let $\sigma_t^2(z) := k_t(z, z)$ represent the posterior variance of the model. Define $\mu_0 := 0$ and $\sigma_0^2(z) := k(z, z)$.

The maximum information gain over $t$ rounds is defined as [1]

$$\gamma_t := \max_{A \subset \mathcal{X} \times \mathcal{C}: |A| = t} \frac{1}{2} \log(\det(\boldsymbol{I}_t + \lambda^{-1}\boldsymbol{K}_A)) , \tag{6}$$

where $\boldsymbol{K}_A$ is the kernel matrix of the sampling points $A$. The following lemma from [34] which is based on [35, Theorem 2] provides tight confidence intervals for functions $f$ with bounded Hilbert norm in RKHS.

**Lemma 1.** *[34, Theorem 1] Let $\delta \in (0, 1)$, $\bar{\lambda} := \max\{1, \lambda\}$, and*

$$\beta_t(\delta) := \sigma\sqrt{\log(\det(\boldsymbol{K}_{t-1} + \bar{\lambda}\boldsymbol{I}_{t-1})) + 2\log\left(\frac{1}{\delta}\right)} + B .$$

*Then, the following holds with probability at least $1 - \delta$:*
$$|\mu_t(x, c) - f(x, c)| \leq \beta_t(\delta)\sigma_t(x, c), \ \forall x \in \mathcal{X}, \ \forall c \in \mathcal{C}, \ \forall t \geq 1 .$$

For simplicity, we set $\lambda = 1$ in the rest of the paper, and observe that $\log(\det(\boldsymbol{K}_{t-1} + \boldsymbol{I}_{t-1})) \leq 2\gamma_{t-1}$. When the confidence parameter $\delta$ is clear from the context, we use $\beta_t$ to represent $\beta_t(\delta)$ to reduce clutter. We define *upper confidence bound* (UCB) and *lower confidence bound* (LCB) for $(x, c) \in \mathcal{X} \times \mathcal{C}$ as follows:

$$\mathrm{ucb}_t(x, c) := \mu_t(x, c) + \beta_t\sigma_t(x, c), \quad \mathrm{lcb}_t(x, c) := \mu_t(x, c) - \beta_t\sigma_t(x, c) .$$

For $x \in \mathcal{X}$, we denote the corresponding UCB and LCB vectors in $\mathbb{R}^n$ by $\mathrm{ucb}_x^t := [\mathrm{ucb}_t(x, c_1), \ldots, \mathrm{ucb}_t(x, c_n)]^\mathsf{T}$ and $\mathrm{lcb}_x^t := [\mathrm{lcb}_t(x, c_1), \ldots, \mathrm{lcb}_t(x, c_n)]^\mathsf{T}$. Also let $f_x := [f(x, c_1), \ldots, f(x, c_n)]^\mathsf{T}$.

# 3 RoBOS for robust Bayesian satisficing

To perform *robust Bayesian satisficing* (RBS), we propose a learning algorithm called *Robust Bayesian Optimistic Satisficing* (RoBOS), whose pseudocode is given in Algorithm 1. At the beginning of each round $t$, RoBOS observes the reference distribution $w_t$. Then, it computes UCB index $\text{ucb}_t(x, c)$ for each action-context pair $(x, c) \in \mathcal{X} \times \mathcal{C}$, by using the GP posterior mean and standard deviation at round $t$. UCB indices, $\tau$ and $w_t$ are used to compute the *estimated fragility* of each action $x$, at round $t$, given as

$$\hat{\kappa}_{\tau,t}(x) = \begin{cases} \max_{w \in \Delta(\mathcal{C}) \setminus \{w_t\}} \frac{\tau - \langle w, \text{ucb}_x^t \rangle}{\|w - w_t\|_M} & \text{if } \langle w_t, \text{ucb}_x^t \rangle \geq \tau \\ +\infty & \text{if } \langle w_t, \text{ucb}_x^t \rangle < \tau \end{cases} \tag{7}$$

where $\Delta(\mathcal{C})$ represents the probability simplex over $\mathcal{C}$, $M$ represents the $n$ by $n$ kernel matrix and $\|w\|_M := \sqrt{w^\mathsf{T} M w}$ is the MMD measure. Specifically, given the kernel $k_\mathcal{C} : \mathcal{C} \times \mathcal{C} \to \mathbb{R}_+$, $M$ is the kernel matrix of the context set $\mathcal{C}$, i.e., $M_{ij} = k_\mathcal{C}(c_i, c_j)$. The estimated fragility $\hat{\kappa}_{\tau,t}(x)$ of an action $x$, is an optimistic proxy for the true fragility $\kappa_{\tau,t}(x)$. Note that when $\hat{\kappa}_{\tau,t}(x) \leq 0$, the threshold $\tau$ is achieved under any context distribution given the UCBs of rewards of $x$. On the other hand, if $\tau > \mathbb{E}_{c \sim P_t}[\text{ucb}_t(x, c)]$, then $\tau$ cannot be achieved under the reference distribution given the UCB indices, thus $\hat{\kappa}_{\tau,t}(x) = \infty$. The next lemma, whose proof is given in the appendix, relates $\hat{\kappa}_{\tau,t}(x)$ with $\kappa_{\tau,t}(x)$.

**Lemma 2.** *Fix $\tau \in \mathbb{R}$. With probability at least $1 - \delta$, $\hat{\kappa}_{\tau,t}(x) \leq \kappa_{\tau,t}(x)$ for all $x \in \mathcal{X}$ and $t \geq 1$.*

To perform robust satisficing, RoBOS chooses as $x_t$ the action with the lowest estimated fragility, i.e., $x_t = \arg\min_{x \in \mathcal{X}} \hat{\kappa}_{\tau,t}(x)$. After action selection, $c_t$ and $y_t$ are observed from the environment, which are then used to update the GP posterior.

---

**Algorithm 1:** RoBOS

---

Inputs $\mathcal{X}, \mathcal{C}, \tau$, GP kernel $k$, $\mu_0 = 0$, $\sigma$, confidence parameter $\delta$, $B$
**for** $t = 1, 2, \ldots$ **do**
    1. Observe the reference distribution $w_t$
    2. Compute $\text{ucb}_t(x, c) = \mu_t(x, c) + \beta_t \sigma_t(x, c)$ for all $x \in \mathcal{X}$ and $c \in \mathcal{C}$
    3. Compute $\hat{\kappa}_{\tau,t}(x)$ as in (7)
    4. Choose action $x_t = \arg\min_{x \in \mathcal{X}} \hat{\kappa}_{\tau,t}(x)$
    5. Observe $c_t \sim P_t^*$ and $y_t = f(x_t, c_t) + \eta_t$
    6. Use $\{x_t, c_t, y_t\}$ to compute $\mu_{t+1}$ and $\sigma_{t+1}$
**end**

---

# 4 Regret analysis

We will analyze the regret under the assumption that (1) is feasible.

**Assumption 1.** *Let $\hat{x}_t := \arg\max_{x \in \mathcal{X}} \langle w_t, f_x \rangle$ be the best action under the reference distribution. We assume that $\tau \leq \langle w_t, f_{\hat{x}_t} \rangle$ for all $t \in [T]$, i.e., the threshold is at most the solution to the SO problem.*

If Assumption 1 does not hold in round $t$, then (1) is infeasible, which means that $\kappa_{\tau,t} = \infty$ and there is no robust satisficing solution. Therefore, measuring the regret in such a round will be meaningless. In practice, if the learner is flexible about its aspiration level, Assumption 1 can be relaxed by dynamically selecting $\tau$ at each round to be less than $\langle w_t, \text{lcb}_{\hat{x}_t'}^t \rangle$, where $\hat{x}_t' := \arg\max_{x \in \mathcal{X}} \langle w_t, \text{lcb}_x^t \rangle$.[3]

The following theorem provides an upper bound on the robust satisficing regret of RoBOS based on the maximum information gain given in (6).

**Theorem 3.** *Fix $\delta \in (0, 1)$. When RoBOS is run under Assumption 1 with confidence parameter $\delta/2$ and $\beta_t := \beta_t(\delta/2)$, where $\beta_t(\delta)$ is defined in Lemma 1, then with a probability of at least $1 - \delta$, the*

---

[3]Indeed, our algorithm can be straightforwardly adapted to work with dynamic thresholds $\tau_t$, $t \geq 1$. When we also change the thresholds in our regret definitions (4) and (5) to $\tau_t$, and update Assumption 1 such that $\tau_t \leq \langle w_t, f_{\hat{x}_t} \rangle$, $t \in [T]$, all derived regret bounds still hold.

*robust satisficing regret $R_T^{rs}$ of RoBOS is upper-bounded as follows:*

$$R_T^{rs} \leq 4\beta_T \sqrt{T\left(2\gamma_T + 2\log\left(\frac{12}{\delta}\right)\right)}.$$

*Proof Sketch:* Assume that confidence bounds in Lemma 1 hold (happens with probability at least $1 - \delta/2$). Let $r_t^{rs} := \tau - \kappa_{\tau,t}\Delta(P_t^*, P_t) - \mathbb{E}_{c \sim P_t^*}[f(x_t, c)]$ be the *instantaneous regret*. Robust satisficing regret can be written as $R_T^{rs} = \sum_{t=1}^{T} r_t^{rs}\mathbb{I}(r_t^{rs} \geq 0)$. We have

$$r_t^{rs} \leq \tau - \kappa_{\tau,t}\|w_t^* - w_t\|_M - \langle w_t^*, \mathrm{ucb}_{x_t}^t\rangle + 2\beta_t\langle w_t^*, \sigma_t(x_t, \cdot)\rangle \tag{8}$$

$$\leq \tau - \kappa_{\tau,t}\|w_t^* - w_t\|_M - (\tau - \hat{\kappa}_{\tau,t}\|w_t^* - w_t\|_M) + 2\beta_t\langle w_t^*, \sigma_t(x_t, \cdot)\rangle \tag{9}$$

$$= \|w_t^* - w_t\|_M(\hat{\kappa}_{\tau,t} - \kappa_{\tau,t}) + 2\beta_t\langle w_t^*, \sigma_t(x_t, \cdot)\rangle$$

$$\leq 2\beta_t\langle w_t^*, \sigma_t(x_t, \cdot)\rangle,$$

where (8) comes from the confidence bounds in Lemma 1, (9) comes from the fact that our algorithm at each round guarantees $\langle w_t^*, \mathrm{ucb}_{x_t}^t\rangle \geq \tau - \hat{\kappa}_{\tau,t}\|w_t^* - w_t\|_M$, and the last inequality is due to Lemma 2. The rest of the proof follows the standard methods used in the Gaussian process literature together with an auxiliary concentration lemma. □

Our analysis uses the fact that under Assumption 1, both the fragility $\kappa_{\tau,t}$ and the estimated fragility $\hat{\kappa}_{\tau,t}$ are finite. We also note that when $\tau > \langle w_t, f_{\hat{x}_t}\rangle$ and $P_t \neq P_t^*$, then $\kappa_{\tau,t} = \infty$, and the instantaneous regret is $(\tau - \kappa_{\tau,t}\Delta(P_t^*, P_t) - \mathbb{E}_{c \sim P_t^*}[f(x_t, c)])^+ = 0$. Setting $\beta_T$ as in Lemma 1, Theorem 3 gives a regret bound of $\tilde{O}(\gamma_T\sqrt{T})$. The next corollary uses the bounds on $\gamma_T$ from [36] to bound the regret for known kernel families.

**Corollary 1.** *When the kernel $k(z, z')$ is either Mátern-$\nu$ kernel or squared exponential kernel, the robust satisficing regret of RoBOS, on an input domain of dimension $d$, is bounded by: $\tilde{O}(T^{\frac{2\nu+3d}{4\nu+2d}})$ and $\tilde{O}(\sqrt{T})$ respectively.*

Next, we analyze the lenient regret of RoBOS. As we discussed in Section 2, lenient regret is a stronger regret measure under which even $x_t^*$ can suffer linear regret. Therefore, our lenient regret bound depends on the amount of distribution shift $\epsilon_t := \|w_t^* - w_t\|_M, t \in [T]$. We highlight that regret analysis of RoBOS is different than that of DRBO in [10], which is done under the assumption that $\|w_t^* - w_t\|_M \leq \epsilon_t'$, for some $\epsilon_t' \geq 0$, $t \in [T]$. The major difference is that DRBO requires $(\epsilon_t')_{t \in [T]}$ as input while RoBOS does not require $(\epsilon_t)_{t \in [T]}$ as input. Also note that $\epsilon_t \leq \epsilon_t'$.

Before stating the lenient regret bound, we let $B' := \max_x \|f_x\|_{M^{-1}}$. It is known that $B' \leq B\sqrt{\lambda_{\max}(M^{-1})n}$, where $\lambda_{\max}(M^{-1})$ represents the largest eigenvalue of $M^{-1}$ and $B$ represents an upper bound on the RKHS norm of $f$ [10].

**Theorem 4.** *Fix $\delta \in (0, 1)$. Under Assumption 1, when RoBOS is run with confidence parameter $\delta/2$ with $\beta_t := \beta_t(\delta/2)$, where $\beta_t$ is defined in Lemma 1, then with a probability of at least $1 - \delta$, the lenient regret $R_T^l$ of RoBOS is upper-bounded as follows:*

$$R_T^l \leq 4\beta_T\sqrt{T\left(2\gamma_T + 2\log\left(\frac{12}{\delta}\right)\right)} + B'\sum_{t=1}^{T}\epsilon_t.$$

*Proof Sketch:* When $\langle w_t, \mathrm{ucb}_x^t\rangle \geq \tau$, let $\bar{w}_x^t := \arg\max_{w \in \Delta(\mathcal{C})\setminus\{w_t\}} \frac{\tau - \langle w, \mathrm{ucb}_x^t\rangle}{\|w - w_t\|_M}$ be the context distribution that achieves $\hat{\kappa}_{\tau,t}(x)$. We have

$$r_t^l := \tau - \langle w_t^*, f_{x_t}\rangle$$

$$= \tau - \langle w_t^*, \mathrm{ucb}_{x_t}^t\rangle + \langle w_t^*, \mathrm{ucb}_{x_t}^t - f_{x_t}\rangle$$

$$\leq \tau - \langle w_t^*, \mathrm{ucb}_{x_t}^t\rangle + 2\beta_t\langle w_t^*, \sigma_t(x_t, \cdot)\rangle, \tag{10}$$

where (10) follows from the confidence bounds in Lemma 1.

Consider $w_t^* = w_t$. By Assumption 1 and the selection rule of RoBOS (i.e., $\langle w_t, \mathrm{ucb}_{x_t}^t\rangle \geq \tau$), we obtain

$$r_t^l \leq \tau - \langle w_t, \mathrm{ucb}_{x_t}^t\rangle + 2\beta_t\langle w_t^*, \sigma_t(x_t, \cdot)\rangle \leq 2\beta_t\langle w_t^*, \sigma_t(x_t, \cdot)\rangle.$$

Consider $w_t^* \neq w_t$. Continuing from (10)

$$r_t^l \leq \|w_t^* - w_t\|_M \frac{\tau - \langle w_t^*, \mathrm{ucb}_{x_t}^t \rangle}{\|w_t^* - w_t\|_M} + 2\beta_t \langle w_t^*, \sigma_t(x_t, \cdot) \rangle$$

$$\leq \|w_t^* - w_t\|_M \hat{\kappa}_{\tau,t}(x_t) + 2\beta_t \langle w_t^*, \sigma_t(x_t, \cdot) \rangle \tag{11}$$

$$\leq \|w_t^* - w_t\|_M \hat{\kappa}_{\tau,t}(\hat{x}_t) + 2\beta_t \langle w_t^*, \sigma_t(x_t, \cdot) \rangle \tag{12}$$

$$\leq \|w_t^* - w_t\|_M \frac{\langle w_t, f_{\hat{x}_t} \rangle - \langle \bar{w}_{\hat{x}_t}^t, \mathrm{ucb}_{\hat{x}_t}^t \rangle}{\|\bar{w}_{\hat{x}_t}^t - w_t\|_M} + 2\beta_t \langle w_t^*, \sigma_t(x_t, \cdot) \rangle \tag{13}$$

$$= \|w_t^* - w_t\|_M \|f_{\hat{x}_t}\|_{M^{-1}} + 2\beta_t \langle w_t^*, \sigma_t(x_t, \cdot) \rangle \tag{14}$$

$$\leq \epsilon_t B' + 2\beta_t \langle w_t^*, \sigma_t(x_t, \cdot) \rangle\,, \tag{15}$$

where (11) comes from the definition of $\hat{\kappa}_{\tau,t}(x_t)$; (12) follows from $x_t = \arg\min_{x \in \mathcal{X}} \hat{\kappa}_{\tau,t}(x)$; (13) results from the assumption on $\tau$ in Assumption 1; (14) utilizes Lemma 1 and the Cauchy-Schwarz inequality; and finally (15) follows from the definition of $\epsilon_t$. The remainder of the proof follows similar arguments as in the proof of Theorem 3. $\qquad \square$

Theorem 4 shows that the lenient regret scales as $\tilde{O}(\gamma_T \sqrt{T} + E_T)$, where $E_T := \sum_{t=1}^T \epsilon_t$. An important special case in which $E_T$ is sublinear is *data-driven optimization*. For this case, $P_t^* = P^*$ is fixed for $t \in [T]$, $P_1$ is the uniform distribution over the contexts, and $P_t = \sum_{s=1}^{t-1} \delta_{c_s}$, $t > 1$ is the empirical distribution of the observed contexts, where $\delta_c$ is the Dirac measure defined for $c \in \mathcal{C}$ such that $\delta_c(A) = 1$ if $c \in A$ and 0 otherwise. The next result follows from steps similar to the proof of [10, Corollary 4].

**Lemma 5.** *Consider data-driven optimization model with $\delta \in (0, 1)$. Under Assumption 1, when RoBOS is run with confidence parameter $\delta/3$ with $\beta_t := \beta_t(\delta/3)$, where $\beta_t$ is defined in Lemma 1, then with a probability of at least $1 - \delta$*

$$R_T^l \leq 4\beta_T \sqrt{T\left(2\gamma_T + 2\log\left(\frac{12}{\delta}\right)\right)} + B'\epsilon_1 + B'2\sqrt{T}\left(2 + \sqrt{2\log\left(\frac{\pi^2 T^2}{2\delta}\right)}\right).$$

## 5   Experiments

We evaluate the proposed algorithm on one synthetic and one real-world environment. We compare the lenient regret and robust satisficing regret of RoBOS with the following benchmark algorithms. Our implementation is available at http://github.com/Bilkent-CYBORG/RoBOS.

*SO*: As the representative of SO we use a stochastic version of the GP-UCB algorithm that samples at each round point $x_t = \arg\max_{x \in \mathcal{X}} \mathbb{E}_{c \sim P_t}[\mathrm{ucb}_t(x, c)]$.
*DRBO*: For the DRO approach we consider the DRBO algorithm from [10]. DRBO, at each round samples $x_t = \arg\max_{x \in \mathcal{X}} \inf_{P \in \mathcal{U}_t} \mathbb{E}_{c \sim P}[\mathrm{ucb}_t(x, c)]$.
*WRBO:* For the WRO approach we consider a maxi-min algorithm we call WRBO, that maximizes the worst-case reward over the context set. Note that when the ambiguity set $\mathcal{U}_t$ of DRBO is $\mathcal{P}_0$, i.e., the set of all possible distributions on $\mathcal{C}$, DRBO reduces to WRBO which samples at each round the point $x_t = \arg\max_{x \in \mathcal{X}} \min_{c \in \mathcal{C}} \mathrm{ucb}_t(x, c)$.

**Synthetic environment.** The objective function is given Figure 1(Left). The reference distribution and the true distribution are chosen to be stationary with $P_t = \mathcal{N}(2, 1)$ and $P_t^* = \mathcal{N}(0, 25)$. We denote the true MMD distance $\Delta(P_t, P_t^*)$ with $\epsilon$, and run different instances of DRBO with ambiguity balls of radius $r = 3\epsilon$, $r = \epsilon$, and $r = \epsilon/3$. When the radius of the ambiguity ball is $\epsilon$, DRBO knows the exact distributional shift. The threshold $\tau$ is set to 60% of the maximum value of the objective function, i.e., $\tau = 0.6 \max_{(x,c) \in \mathcal{X} \times \mathcal{C}} f(x, c) \approx 0.65 Z_t$. We use an RBF kernel with Automatic Relevance Determination (ARD) and lengthscales 0.2 and 5 respectively for dimensions $\mathcal{X}$ and $\mathcal{C}$. Simulations are run with observation noise $\eta_t \sim \mathcal{N}(0, 0.02^2)$.

Results for the robust satisficing and lenient regrets are given in Figure 2. Under this configuration, the RS solution, as noted in Figure 1, is the only solution that achieves the desired threshold $\tau$. Hence, algorithms that converge to a different solution accumulate linear lenient regret, as can be seen in Figure 2(Right). When DRBO is run with an overconfident ambiguity set (radius $\epsilon/3$), it converges to a suboptimal solution (hexagon); when the ambiguity set is underconfident (radius $3\epsilon$), it converges

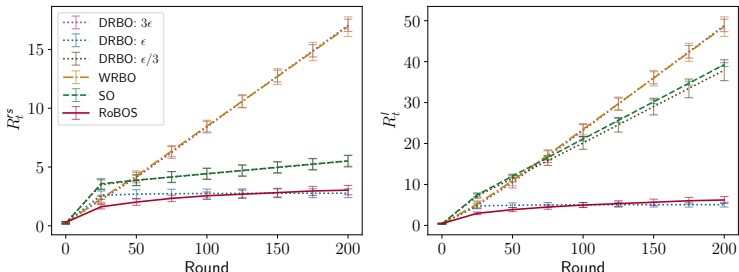

Figure 2: Results for synthetic environment. Plots show robust satisficing regret (left) and lenient regret (right) averaged over 50 independent runs with error bars corresponding to standard deviations divided by 2.

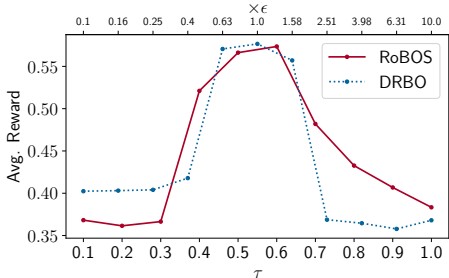

Figure 3: Average reward for different $\tau$ and $r$ values in RoBOS and DRBO. $\tau$ is selected linearly between minimum and maximum function values. The radius $r$ of the ambiguity ball for DRBO is selected from $0.1\epsilon$ to $10\epsilon$. Plots show average of 10 independent runs each with 200 rounds.

to another suboptimal solution (cross). Only when the distribution shift is precisely known and the ambiguity set is adjusted accordingly (radius $\epsilon$), DRBO converges to the RS solution. Refer to Figure 1 to see the exact solutions converged.

We also investigate the sensitivity of the choice of $\tau$ in RoBOS compared to the choice of $r$ in DRBO. Figure 3 compares the average rewards of RoBOS and DRBO for a wide range of $\tau$ and $r$ values. While RoBOS is designed to satisfice the reward rather than to maximize, its performance remains competitive with DRBO across diverse hyperparameter settings. Notably, for $\tau \in [0.1, 0.3]$, RoBOS opts for the solution indicated by a cross in Figure 1, signifying a trade-off: with a smaller $\tau$, RoBOS prioritizes robustness guarantees over maximizing the reward.

**Insulin dosage for T1DM.** We test our algorithm on the problem of personalized insulin dose allocation for Type 1 Diabetes Mellitus (T1DM) patients. We use the open-source implementation of the U.S. FDA-approved University of Virginia (UVA)/PADOVA T1DM simulator [37, 5]. The simulator takes in as input the fasting blood glucose level of the patient, the amount of carbohydrate intake during the monitored meal and the insulin dosage given to the patient, it gives an output of the blood glucose level measured 150 minutes after the meal. We assume the insulin is administered to the patient right after the meal. Similar to [5], we set the target blood glucose level as $K = 112.5$ mg/dl and define the pseudo-reward function as $r(t) = -|o(t) - K|$ where $o(t)$ is the achieved blood glucose level at round $t$. We further define a safe blood glucose level range as 102.5 - 122.5 mg/dl. For our setup, this corresponds to setting the threshold $\tau = -10$. At each round $t$, the environment picks the true and reference distributions as $P_t \sim \mathcal{N}(\zeta_t, 2.25)$ and $P_t^* \sim \mathcal{N}(\zeta_t + N, 9)$ where $\zeta_t \sim \mathcal{U}(20, 80)$ and $N$ is the random term setting the distributional shift. We define the action set $\mathcal{X}$ to be the insulin dose and the context set $\mathcal{C}$ to be the amount of carbohydrate intake. Here the distributional shift can be interpreted as the estimation error of the patient on their carbohydrate intake. We ran our experiments with $N \sim \mathcal{U}(-6, 6)$ and with $N_t \sim \mathcal{U}(-6/\log(t+2), 6/\log(t+2))$. Simulations are run with observation noise $\eta_t \sim \mathcal{N}(0, 1)$. For the GP surrogate, we used a Matérn-$\nu$ kernel with length-scale parameter 10. As seen in Figures 4a and 4b, when the amount of distribution

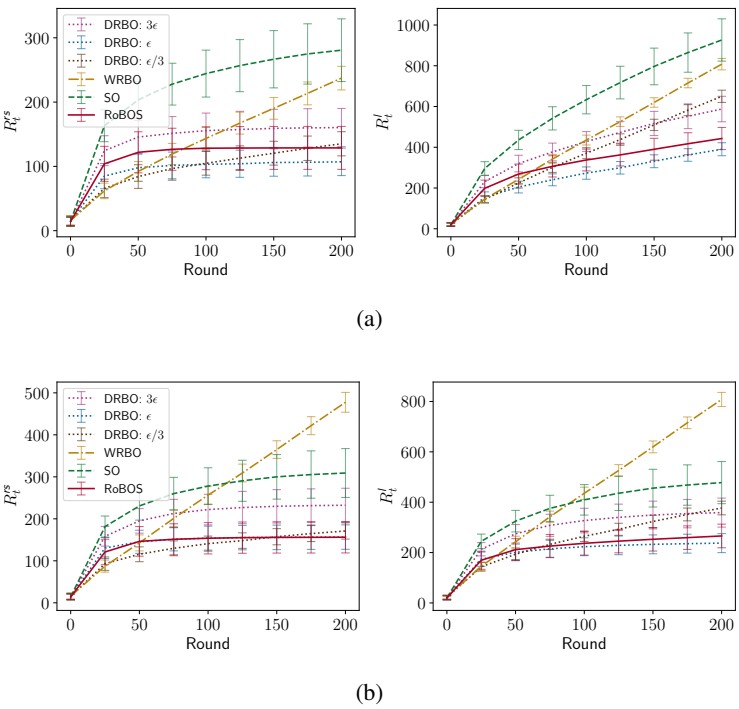

Figure 4: Results for insulin dose allocation simulation where $\epsilon_t$ is the true MMD distance between the true and reference distributions at round $t$, and the threshold $\tau = -10$ specifies a satisficing threshold of 10 mg/dl away from the target blood glucose level. On (b), the true MMD distance $\epsilon_t \approx \epsilon_0 / \log(t)$ decays with $t$. Plots show robust satisficing regret (Left) and lenient regret (Right) averaged over 50 independent runs with error bars corresponding to standard deviations divided by 2.

shift at each round is known exactly by DRBO, it can perform better than RoBOS. However, when the distributional shift is either underestimated or overestimated, RoBOS achieves better results. Plots of the average cumulative rewards of the algorithms can be found in the appendix.

## 6   Conclusion, limitations, and future research

We introduced *robust Bayesian satisficing* as a new sequential decision-making paradigm that offers a satisfactory level of protection against incalculable distribution shifts. We introduced the first RBS algorithm RoBOS, and proved information gain based lenient and robust satisficing regret bounds.

In our experiments, we observed that when the range of the distribution shift can be correctly estimated with a tight uncertainty set $\mathcal{U}_t$ centered at the reference distribution, DRBO [10] can perform better than RoBOS, especially when it comes to maximizing total reward. This is not unexpected since one can guarantee better performance with more information about the distribution shift. Nevertheless, in cases when the estimated shift does not adequately represent the true shift or when the main objective is to achieve a desired value instead of maximizing the reward, RoBOS emerges as a robust alternative.

Our fundamental research on RBS brings forth many interesting future research directions. One potential direction is to extend RoBOS to work in continuous action and context spaces. This will require a more nuanced regret analysis and computationally efficient procedures to calculate the estimated fragility at each round. Another interesting future work is to design alternative acquisition strategies for RBS. For instance, one can investigate a Thompson sampling based approach instead of the UCB approach we pursued in this work.

**Acknowledgements:** Y. Cahit Yıldırım was supported by Turk Telekom as part of 5G and Beyond Joint Graduate Support Programme coordinated by Information and Communication Technologies Authority.

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

# A    Table of notations

Table 2: Table of notations

| Notation | Description |
|---|---|
| $\tau$ | aspiration level (threshold) |
| $\|w\|_M$ | $\sqrt{w^{\mathsf{T}} M w}$ MMD measure with kernel matrix $M$ |
| $\mathcal{X}$ | Action set |
| $\mathcal{C}$ | Context set |
| $w_t^*$ | True distribution at round $t$ |
| $w_t$ | Reference distribution at round $t$ |
| $f_x$ | $[f(x, c_1), \ldots, f(x, c_n)]^{\mathsf{T}}$ |
| $\mathrm{ucb}_x^t$ | upper confidence bound of $f_x$ given by $[\mathrm{ucb}_t(x, c_1), \ldots, \mathrm{ucb}_t(x, c_n)]^{\mathsf{T}}$ |
| $\kappa_{\tau,t}(x)$ | $\begin{cases} \max\left\{\max_{w \in \Delta(\mathcal{C}) \setminus \{w_t\}} \frac{\tau - \langle w, f_x \rangle}{\|w - w_t\|_M}, 0\right\} & \text{if } \langle w_t, f_x \rangle \geq \tau \\ +\infty & \text{if } \langle w_t, f_x \rangle < \tau \end{cases}$ |
| $\hat{\kappa}_{\tau,t}(x)$ | $\begin{cases} \max_{w \in \Delta(\mathcal{C}) \setminus \{w_t\}} \frac{\tau - \langle w, \mathrm{ucb}_x^t \rangle}{\|w - w_t\|_M} & \text{if } \langle w_t, \mathrm{ucb}_x^t \rangle \geq \tau \\ +\infty & \text{if } \langle w_t, \mathrm{ucb}_x^t \rangle < \tau \,. \end{cases}$ |
| $\hat{x}_t$ | $\arg\max_{x \in \mathcal{X}} \langle w_t, f_x \rangle$ |
| $x_t^*$ | $\arg\min_{x \in \mathcal{X}} \kappa_{\tau,t}(x)$ when $\langle w_t, f_{\hat{x}_t} \rangle \geq \tau$ |
| $x_t$ | $\arg\min_{x \in \mathcal{X}} \hat{\kappa}_{\tau,t}(x)$ when $\langle w_t, f_{\hat{x}_t} \rangle \geq \tau$ |
| $\kappa_{\tau,t}$ | $\kappa_{\tau,t}(x_t^*)$ when $\langle w_t, f_{\hat{x}_t} \rangle \geq \tau$ |
| $\hat{\kappa}_{\tau,t}$ | $\hat{\kappa}_{\tau,t}(x_t)$ when $\langle w_t, f_{\hat{x}_t} \rangle \geq \tau$ |
| $\bar{\bar{w}}_x^t$ | $\arg\max_{w \in \Delta(\mathcal{C}) \setminus \{w_t\}} \frac{\tau - \langle w, f_x \rangle}{\|w - w_t\|_M}$ when $\langle w_t, f_x \rangle \geq \tau$ |
| $\bar{w}_x^t$ | $\arg\max_{w \in \Delta(\mathcal{C}) \setminus \{w_t\}} \frac{\tau - \langle w, \mathrm{ucb}_x^t \rangle}{\|w - w_t\|_M}$ when $\langle w_t, \mathrm{ucb}_x^t \rangle \geq \tau$ |

# B    Additional experimental results

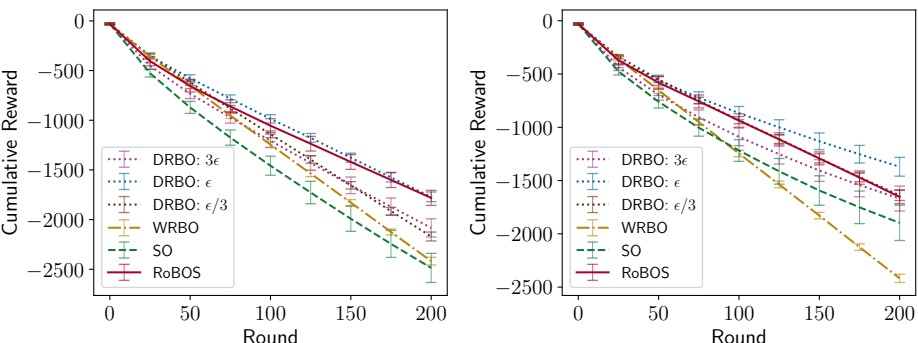

Figure 5: Cumulative rewards of the insulin dosage for T1DM experiments given in the main paper. Left and right plots are continuations of Figures 4a and 4b, respectively. Plots show average of 50 independent runs with error bars corresponding to standard deviation divided by 2. The pseudo-reward function is defined as $r(t) = -|o(t) - K|$.

# C    Proofs

### C.1    Proof of Lemma 2

Assume that the confidence intervals in Lemma 1 hold. When $\hat{\kappa}_{\tau,t}(x) = \infty$, we also have $\kappa_{\tau,t}(x) = \infty$. When $\hat{\kappa}_{\tau,t}(x) < \infty$ and $\kappa_{\tau,t}(x) = \infty$, the inequality holds.

Recall that when $\langle w_t, \mathrm{ucb}_x^t \rangle \geq \tau$, $\bar{w}_x^t := \arg\max_{w \in \Delta(\mathcal{C}) \backslash \{w_t\}} \frac{\tau - \langle w, \mathrm{ucb}_x^t \rangle}{\|w - w_t\|_M}$ represents the context distribution that achieves $\hat{\kappa}_{\tau,t}(x)$. When $\langle w_t, f_x \rangle \geq \tau$, let $\bar{\bar{w}}_x^t := \arg\max_{w \in \Delta(\mathcal{C}) \backslash \{w_t\}} \frac{\tau - \langle w, f_x \rangle}{\|w - w_t\|_M}$ be the context distribution that achieves $\kappa_{\tau,t}(x)$. When $\hat{\kappa}_{\tau,t}(x) < \infty$ and $\kappa_{\tau,t}(x) < \infty$, we have

$$
\begin{aligned}
\hat{\kappa}_{\tau,t}(x) - \kappa_{\tau,t}(x) &\leq \frac{\tau - \langle \bar{w}_x^t, \mathrm{ucb}_x^t \rangle}{\|\bar{w}_x^t - w_t\|_M} - \frac{\tau - \langle \bar{\bar{w}}_x^t, f_x \rangle}{\|\bar{\bar{w}}_x^t - w_t\|_M} \\
&\leq \frac{\tau - \langle \bar{w}_x^t, \mathrm{ucb}_x^t \rangle}{\|\bar{w}_x^t - w_t\|_M} - \frac{\tau - \langle \bar{w}_x^t, f_x \rangle}{\|\bar{w}_x^t - w_t\|_M} \\
&= \frac{\langle \bar{w}_x^t, f_x - \mathrm{ucb}_x^t \rangle}{\|\bar{w}_x^t - w_t\|_M} \\
&\leq 0 \, .
\end{aligned}
$$

## C.2 An Auxiliary Concentration Lemma

**Lemma 6.** *([10, Lemma 7]) Let $S_t \geq 0$ be a non-negative stochastic process with filtration $\mathcal{F}_t$, and define $m_t = \mathbb{E}[S_t | \mathcal{F}_{t-1}]$. Further assume that $S_t \leq B$ for $B \geq 1$. Then for any $T \geq 1$, with probability at least $1 - \delta$ it holds that*

$$
\begin{aligned}
\sum_{t=1}^T m_t &\leq 2 \sum_{t=1}^T S_t + 4B \log \frac{1}{\delta} + 8B \log(4B) + 1 \\
&\leq 2 \sum_{t=1}^T S_t + 8B \log \frac{6B}{\delta} \, .
\end{aligned}
$$

## C.3 Proof of Theorem 3

The robust satisficing regret is upper bounded by the sum of instantaneous regrets $r_t^{rs} := \tau - \kappa_{\tau,t} \|w_t^* - w_t\|_M - \langle w_t^*, f_{x_t} \rangle$ as follows:

$$
\begin{aligned}
R_T^{rs} &= \sum_{t=1}^T \left( \tau - \kappa_{\tau,t} \Delta(P_t^*, P_t) - \mathbb{E}_{c \sim P_t^*}[f(x_t, c)] \right)^+ \\
&= \sum_{t=1}^T (\tau - \kappa_{\tau,t} \|w_t^* - w_t\|_M - \langle w_t^*, f_{x_t} \rangle)^+ \\
&= \sum_{t=1}^T r_t^{rs} \mathbb{I}(r_t^{rs} \geq 0) \, .
\end{aligned}
\tag{16}
$$

Assume that the confidence intervals in Lemma 1 hold (an event that happens with probability at least $1 - \delta/2$). For the instantaneous regret we have

$$
\begin{aligned}
r_t^{rs} &= \tau - \kappa_{\tau,t} \|w_t^* - w_t\|_M - \langle w_t^*, \mathrm{ucb}_{x_t}^t \rangle + \langle w_t^*, \mathrm{ucb}_{x_t}^t - f_{x_t} \rangle \\
&\leq \tau - \kappa_{\tau,t} \|w_t^* - w_t\|_M - \langle w_t^*, \mathrm{ucb}_{x_t}^t \rangle + 2\beta_t \langle w_t^*, \sigma_t(x_t, \cdot) \rangle \tag{17} \\
&\leq \tau - \kappa_{\tau,t} \|w_t^* - w_t\|_M - (\tau - \hat{\kappa}_{\tau,t} \|w_t^* - w_t\|_M) + 2\beta_t \langle w_t^*, \sigma_t(x_t, \cdot) \rangle \tag{18} \\
&= \|w_t^* - w_t\|_M (\hat{\kappa}_{\tau,t} - \kappa_{\tau,t}) + 2\beta_t \langle w_t^*, \sigma_t(x_t, \cdot) \rangle \\
&\leq 2\beta_t \langle w_t^*, \sigma_t(x_t, \cdot) \rangle \, . \tag{19}
\end{aligned}
$$

In the derivation above, (17) follows from the confidence bounds given in Lemma 1; (18) follows from $\langle w_t^*, \mathrm{ucb}_{x_t}^t \rangle \geq \tau - \hat{\kappa}_{\tau,t}(x_t) \|w_t^* - w_t\|_M$ and $\hat{\kappa}_{\tau,t} = \hat{\kappa}_{\tau,t}(x_t)$; (19) is due to Lemma 2.

From this point on, we bound the robust satisficing regret by following standard steps for bounding regret of GP bandits (see e.g., [10]). First, by plugging the upper bound on $r_t^{rs}$ obtained in (19) to

(16), and using monotonicity of $\beta_t$, we obtain

$$R_T^{rs} \leq 2\beta_T \sum_{t=1}^{T} \langle w_t^*, \sigma_t(x_t, \cdot) \rangle \tag{20}$$

$$\leq 2\beta_T \sqrt{T \sum_{t=1}^{T} \langle w_t^*, \sigma_t(x_t, \cdot) \rangle^2} \tag{21}$$

$$\leq 2\beta_T \sqrt{T \sum_{t=1}^{T} \langle w_t^*, \sigma_t(x_t, \cdot)^2 \rangle}, \tag{22}$$

where (21) uses the Cauchy-Schwarz inequality and (22) uses the Jensen inequality.

To complete the proof we use Lemma 6 to relate the expectation of the posterior variance in (22) to the posterior variance of the observations. Namely, we have with probability at least $1 - \delta/2$

$$\sum_{t=1}^{T} \langle w_t^*, \sigma_t(x_t, \cdot)^2 \rangle \leq 2 \sum_{t=1}^{T} \sigma_t(x_t, c_t)^2 + 8 \log \frac{12}{\delta}. \tag{23}$$

Next, to relate the sum of variances that appears in (23) to the maximum information gain. We use $x \leq 2\log(1 + x)$ for all $x \in [0, 1]$ to obtain

$$\sum_{t=1}^{T} \sigma_t(x_t, c_t)^2 \leq \sum_{t=1}^{T} 2\log(1 + \sigma_t(x_t, c_t)^2)$$

$$\leq 4\gamma_T, \tag{24}$$

where (24) follows from [35, Lemma 3].

Finally, to bound the robust satisficing regret, we plug the upper bound in (24) to the r.h.s. of (23), and use it to upper bound (22) as follows

$$R_T^{rs} \leq 4\beta_T \sqrt{T \left( 2\gamma_T + 2\log\left(\frac{12}{\delta}\right) \right)}.$$

Let $A$ and $B$ be the events that Lemma 1 and Lemma 6 hold. Setting the confidence of each event to $1 - \delta/2$, we can bound from below the probability that both events hold

$$P(A \cap B) = 1 - P(\bar{A} \cup \bar{B}) \geq 1 - P(\bar{A}) - P(\bar{B}) = 1 - \delta,$$

completing the proof of Theorem 3.

### C.4 Proof of Theorem 4

Define the instantaneous regret as $r_t^l := \tau - \langle w_t^*, f_{x_t} \rangle$. We have

$$R_T^l := \sum_{t=1}^{T} \left( \tau - \mathbb{E}_{c \sim P_t^*}[f(x_t, c)] \right)^+ = \sum_{t=1}^{T} r_t^l \mathbb{I}(r_t^l \geq 0). \tag{25}$$

Assume that the confidence intervals in Lemma 1 hold (an event that happens with probability at least $1 - \delta/2$). Then, for the instantaneous regret we have

$$r_t^l = \tau - \langle w_t^*, \mathrm{ucb}_{x_t}^t \rangle + \langle w_t^*, \mathrm{ucb}_{x_t}^t - f_{x_t} \rangle$$

$$\leq \tau - \langle w_t^*, \mathrm{ucb}_{x_t}^t \rangle + 2\beta_t \langle w_t^*, \sigma_t(x_t, \cdot) \rangle. \tag{26}$$

When, $w_t^* = w_t$, by Assumption 1 and the selection rule of RoBOS (i.e., $\langle w_t, \mathrm{ucb}_{x_t}^t \rangle \geq \tau$), we obtain

$$r_t^l \leq \tau - \langle w_t, \mathrm{ucb}_{x_t}^t \rangle + 2\beta_t \langle w_t^*, \sigma_t(x_t, \cdot) \rangle \leq 2\beta_t \langle w_t^*, \sigma_t(x_t, \cdot) \rangle.$$

When $w_t^* \neq w_t$, continuing from (26), we obtain

$$r_t^l \leq \|w_t^* - w_t\|_M \frac{\tau - \langle w_t^*, \mathrm{ucb}_{x_t}^t \rangle}{\|w_t^* - w_t\|_M} + 2\beta_t \langle w_t^*, \sigma_t(x_t, \cdot) \rangle$$

$$\leq \|w_t^* - w_t\|_M \frac{\tau - \langle \bar{w}_{x_t}^t, \mathrm{ucb}_{x_t}^t \rangle}{\|\bar{w}_{x_t}^t - w_t\|_M} + 2\beta_t \langle w_t^*, \sigma_t(x_t, \cdot) \rangle \tag{27}$$

$$\leq \|w_t^* - w_t\|_M \frac{\tau - \langle \bar{w}_{\hat{x}_t}^t, \mathrm{ucb}_{\hat{x}_t}^t \rangle}{\|\bar{w}_{\hat{x}_t}^t - w_t\|_M} + 2\beta_t \langle w_t^*, \sigma_t(x_t, \cdot) \rangle \tag{28}$$

$$\leq \|w_t^* - w_t\|_M \frac{\langle w_t, f_{\hat{x}_t} \rangle - \langle \bar{w}_{\hat{x}_t}^t, \mathrm{ucb}_{\hat{x}_t}^t \rangle}{\|\bar{w}_{\hat{x}_t}^t - w_t\|_M} + 2\beta_t \langle w_t^*, \sigma_t(x_t, \cdot) \rangle \tag{29}$$

$$\leq \|w_t^* - w_t\|_M \frac{\langle w_t - \bar{w}_{\hat{x}_t}^t, f_{\hat{x}_t} \rangle}{\|\bar{w}_{\hat{x}_t}^t - w_t\|_M} + 2\beta_t \langle w_t^*, \sigma_t(x_t, \cdot) \rangle \tag{30}$$

$$\leq \|w_t^* - w_t\|_M \frac{\|\bar{w}_{\hat{x}_t}^t - w_t\|_M \|f_{\hat{x}_t}\|_{M^{-1}}}{\|\bar{w}_{\hat{x}_t}^t - w_t\|_M} + 2\beta_t \langle w_t^*, \sigma_t(x_t, \cdot) \rangle \tag{31}$$

$$= \|w_t^* - w_t\|_M \|f_{\hat{x}_t}\|_{M^{-1}} + 2\beta_t \langle w_t^*, \sigma_t(x_t, \cdot) \rangle$$

$$\leq 2\beta_t \langle w_t^*, \sigma_t(x_t, \cdot) \rangle + \epsilon_t B' \ . \tag{32}$$

Since the final bound is non-negative, it can be used to bound all terms $r_t^l \mathbb{I}(r_t^l \geq 0)$ in $R_T^l$. In the derivation above, (26) follows from Lemma 1; (27) uses the fact that $\bar{w}_{x_t}^t$ is the distribution that achieves $\hat{\kappa}_{\tau,t}(x_t)$; (28) uses the fact that $x_t$ minimizes $\hat{\kappa}_{\tau,t}(x)$, (29) uses Assumption 1, (30) again uses Lemma 1, (31) follows from the Cauchy-Schwarz inequality; and (32) follows from the facts $\|w_t^* - w_t\|_M = \epsilon_t$ and $B' = \max_x \|f_x\|_{M^{-1}}$. From this point on, the regret bound follows the same arguments as in the proof of Theorem 3. In particular, using (32), we write

$$R_T^l \leq 2\beta_T \sum_{t=1}^T \langle w_t^*, \sigma_t(x_t, \cdot) \rangle + B' \sum_{t=1}^T \epsilon_t \ . \tag{33}$$

We complete the proof by continuing from (20), by using the same steps as in the proof of Theorem 3 starting from (20), which leads to the regret bound in the statement of Theorem 4.

### C.5 Proof of Lemma 5

The first term of Lemma 5 directly follows from Theorem 4. For the second term, we use the result of [38, Theorem 3.4], which is restated below.

**[38, Theorem 3.4]** Assume $k(c_i, c_j) \leq 1$ for all $c_i, c_j \in \mathcal{C}$. Then, with probability at least $1 - \delta$

$$d(P^*, \hat{P}_t) \leq \frac{1}{\sqrt{t-1}}(2 + \sqrt{2\log(1/\delta)}) \ .$$

To use the lemma above, let $\delta_t = \frac{2\delta}{\pi^2 t^2}$. Then,

$$\mathbb{P}\left( d(P^*, \hat{P}_t) > \frac{1}{\sqrt{t-1}} \left( 2 + \sqrt{2\log\left(\frac{\pi^2 t^2}{2\delta}\right)} \right) \right) \leq \frac{2\delta}{\pi^2 t^2} \ .$$

Using a union bound, we obtain

$$\mathbb{P}\left( \exists t \in [T] : d(P^*, \hat{P}_t) > \frac{1}{\sqrt{t-1}} \left( 2 + \sqrt{2\log\left(\frac{\pi^2 t^2}{2\delta}\right)} \right) \right) \leq \sum_{t=1}^T \frac{2\delta}{\pi^2 t^2} = \frac{2\delta}{\pi^2} \frac{\pi^2}{6} = \frac{\delta}{3} \ .$$

The inequality above implies that

$$\mathbb{P}\left( \forall t \in [T] : d(P^*, \hat{P}_t) > \frac{1}{\sqrt{t-1}} \left( 2 + \sqrt{2\log\left(\frac{\pi^2 t^2}{2\delta}\right)} \right) \right) \leq 1 - \frac{\delta}{3} \ .$$

Since $\epsilon_t \leq \frac{1}{\sqrt{t-1}} \left( 2 + \sqrt{2 \log \left( \frac{\pi^2 t^2}{2\delta} \right)} \right)$, we have with probability at least $1 - \delta$

$$R_T^l \leq 4\beta_T \sqrt{T \left( 2\gamma_T + 2 \log \left( \frac{12}{\delta} \right) \right)} + B' \sum_{t=1}^{T} \epsilon_t .$$

$$\leq 4\beta_T \sqrt{T \left( 2\gamma_T + 2 \log \left( \frac{12}{\delta} \right) \right)} + B'\epsilon_1 + B' \sum_{t=2}^{T} \frac{1}{\sqrt{t-1}} \left( 2 + \sqrt{2 \log \left( \frac{\pi^2 t^2}{2\delta} \right)} \right)$$

$$\leq 4\beta_T \sqrt{T \left( 2\gamma_T + 2 \log \left( \frac{12}{\delta} \right) \right)} + B'\epsilon_1 + B' \sum_{t=2}^{T} \frac{1}{\sqrt{t-1}} \left( 2 + \sqrt{2 \log \left( \frac{\pi^2 T^2}{2\delta} \right)} \right)$$

$$= 4\beta_T \sqrt{T \left( 2\gamma_T + 2 \log \left( \frac{12}{\delta} \right) \right)} + B'\epsilon_1 + B' \sum_{t=1}^{T-1} \frac{1}{\sqrt{t}} \left( 2 + \sqrt{2 \log \left( \frac{\pi^2 T^2}{2\delta} \right)} \right)$$

$$\leq 4\beta_T \sqrt{T \left( 2\gamma_T + 2 \log \left( \frac{12}{\delta} \right) \right)} + B'\epsilon_1 + B'2\sqrt{T} \left( 2 + \sqrt{2 \log \left( \frac{\pi^2 T^2}{2\delta} \right)} \right) .$$

Also, note that

$$\epsilon_1 = \|w^* - w_1\|_M = \|w^* - \frac{1}{n}\|_M \leq \sup_{\{w: \|w\|_1 = 1\}} \|w - \frac{1}{n}\|_M = O(1) .$$

