# OpenReview forum: "Robust Bayesian Satisficing"
_NeurIPS.cc/2023/Conference — NeurIPS 2023 poster_

### Official Review · Reviewer_KF6g · 2023-06-14

**Soundness:** 3 good
**Presentation:** 2 fair
**Contribution:** 3 good
**Rating:** 6
**Confidence:** 3

**Summary:**

The paper proposes robust Bayesian satisficing, a new setting of BO that is similar to distributionally robust BO. Robust Bayesian satisficing aims to achieve a 'good enough' expected value given by some threshold $\tau$ and relaxed by the distribution distance between a reference and a true distribution. The paper proposes 2 new notions of regret, design an algorithm with regret bounds, and empirically compare this algorithm to DRBO and other suitable baselines.

**Strengths:**

1. Robust satisficing as a new optimization objective for BO is interesting and presents an alternative 'good enough' objective along with a relaxation based on distribution distance that is sensible. It enables distributionally robust BO in another way that does not involve uncertainty sets.
2. The new regret definitions make sense, and the proposed algorithm is supported by theoretical guarantees on its performance via regret bounds.
3. The empirical evaluations present support the claim that the proposed algorithm performs well with the proposed regret definitions.
4. Overall I believe that this work has relevance to the community, subject to the issues raised in the Weaknesses section being addressed properly.

**Weaknesses:**

Technical concerns:
1. How is the threshold $\tau$ to be selected in a real world problem? The paper states that it 'can be expressed as a percentage of the SO solution', but when $f$ is unknown, the expected value of the SO solution is unknown as well. From Algorithm 1, $\tau$ is an input prior to any BO rounds. This is an important question to answer, since one of the claimed advantages over DRBO is that there is no need to pick uncertainty sets which may be unknown a priori. But now you have to pick $\tau$ which is also unknown a priori, so it seems that you have replaced one hyperparameter for another. In order to satisfy Assumption 1, it is claimed that $\tau$ can be dynamically selected, but that makes the regret and thus the regret bounds not well-defined, since the regret is a function of a constant $\tau$.  If $\tau$ is to be dynamically selected and learned, then the algorithm and regret bounds should be explicitly written to take this into consideration, instead of claiming in a footnote that it 'can be straightforwardly adapted to work with dynamic thresholds'.
2. The experiments in the main paper are simple synthetic benchmarks which by themselves are not comprehensive enough an empirical evaluation. I see that you have an interesting real world benchmark on insulin dose allocation in the appendix, why is it hidden there without a reference from the main paper? Why aren't WRBO and SO tested on that benchmark?

Clarity issues:
1. The equation below line 102 defining $\kappa_{\tau, t}$ and the preceding sentence does not quite make sense. It seems to me that you do not need that sentence and equation, simply define fragility as in Eq. (1), and then the following sentence makes clear what $x_t^*$ and $\kappa_{\tau, t}$ are.
2. In all the figures, heatmaps are uninterpretable without a colorbar indicating the values that each color corresponds to.
3. Lemma 1 uses the maximum information gain $\gamma$ before it is defined, and includes a definition for determinant when determinant is not used there.

**Questions:**

No additional questions other than those raised in Weaknesses.

**Limitations:**

Yes.

---

> ### Author Rebuttal · Authors · 2023-08-09
>
> We thank the reviewer for their thoughtful comments and valuable insights.
>
>
> ### Weaknesses
> #### Technical Concerns
> - **(...how $\tau$ can be selected in a real-world problem...)** One real-world experiment concerning safe dose allocation for diabetes patients is presented in the supplementary document. Type 1 Diabetes Mellitus (T1DM) patients require bolus insulin doses (id) after meals for postprandial blood glucose (pbg) regulation. One of the most important factors that affect pbg is meal carbohydrate (cho) intake [C]. Let ${\cal X}$ and ${\cal C}$ represent admissible id and cho values. For $x \in {\cal X}$, $c \in {\cal C}$, let $g(x,c)$ represent the corresponding (expected) bpg value. Function $g$ depends on the patient's characteristics and can be regarded as unknown. The main goal of pbg regulation is to keep pbg close to a target level $K$ in order to prevent two potentially life threatening events called hypoglycemia (e.g., pbg $<$ 70 mg/dl) and hyperglycemia (e.g., pbg $>$ 180 mg/dl). This requires $x\_t$ to be chosen judiciously based on current $c\_t$. Patients rely on a method called cho counting to calculate $c\_t$. Often times, this method is prone errors [D]. The reported cho intake $\zeta\_t$ can differ significantly from $c\_t$. In order to use DRO, one needs to identify a range of plausible distributions for cho calculation errors, which is hard calculate and interpret. On the other hand, specifying $\tau$ corresponds to defining an interval of safe pbg values around $K$ (e.g., pbg $=$ 125 mg/dl) that one is content with, which is in line with the standard clinical practice [E]. We will move this experiment from the supplemental document to the main paper. In addition, in response to the comments of reviewer Gzxd and KF6g, we performed new experiments that compare accumulated rewards of RoBOS and other competing benchmarks. Moreover, we also carried out simulations that shows sensitivity of the results to the aspiration level $\tau$ set by the learner. These results can be found in the response pdf.
>
> - **(...$\tau$ expressed as a percentage of SO solution...)** The interpretation that $\tau$ can be chosen as a percentage of the stochastic optimization problem $Z\_t$ pertains to the case when $f$ is known. We discussed this under the optimization objective subsection before the regret definitions. We will clarify this issue in the revised version. Nevertheless, in learning problems in which $f$ is unknown but the value of $Z\_t$ is known, this interpretation can still be used.
>
> - **(...an alternative way of setting $\tau$ when $f$ is unknown...)** For instance, with the power of Bayesian modelling $f$ can be estimated with confidence bounds which can be used to pick a meaningful $\tau$ value. For example, picking $\tau \geq \max\_{\mathcal{X}\times\mathcal{C}} \text{ucb}\_t(x,c)$ almost  certainly sets us to failure ($\tau$ not achievable with high probability). In contrast, picking $\tau \leq \langle w\_t, \text{lcb}^t\_{\hat{x}'\_t}\rangle$, where $\hat{x}'\_t := \arg\max\_{x \in {\cal X}} \langle w\_t, \text{lcb}^t\_x\rangle$, guarantees with probability at least $1-\delta$ that the optimization problem (1) is feasible.
>
> - **(...regret bounds for time-varying $\tau$...)** By going over the original proofs, it can be verified that indeed $\tau$ can be selected dynamically, $\tau\_t$, $t \geq 1$. To see this, we also change the thresholds in our regret definitions to $\tau\_t$, and update Assumption 1 such that $\tau\_t \leq \langle w\_t, f\_{\hat{x}\_t}\rangle$, $t \in [T]$, all derived regret bounds  still hold.
>
> - **(...experiments...)** The real-world experiment is taken to the main paper with the addition of WRBO and SO benchmarks. During the response phase, we also performed new experiments including comparing cumulative rewards of different algorithms on the true distributions and sensitivity analysis w.r.t. $\tau$. These can be found in the response pdf.
>
>
> #### Clarity
> 1) We thank the reviewer for their suggestion and we note that we have slightly updated the definitions to increase clarity. As you suggested, $\kappa\_{\tau,t}$ can be computed in two steps. We wrote the first equation to highlight robust satisficing as a single optimization problem.
> In RS, the objective is to find $x^*\_t \in \mathcal{X}$ that solves in each round $t$
> \begin{equation}
>     \kappa\_{\tau,t} = \min k
>     ~ \text{s.t.} ~ \mathbb{E}\_{c\sim P}[f(x,c)] \geq \tau - k \Delta(P,P\_t), ~ \forall P \in {\cal P}\_0 ~ , x \in {\cal X}, ~ k \geq 0 ~ . \tag{1}
> \end{equation}
> To find $x^*\_t$, we can first compute the *fragility* of $x \in {\cal X}$ as
> \begin{equation*}
>     \kappa\_{\tau,t}(x) = \min k ~ \text{s.t.} ~ \mathbb{E}\_{c\sim P}[f(x,c)] \geq \tau - k \Delta(P,P\_t), ~ \forall P \in {\cal P}\_0, ~ k \geq 0 ~ .
> \end{equation*}
> The robust satisficing objective is feasible when (1) has a solution.
>
> 2) Colorbars added where necessary.
>
> 3) Fixed.

---

> > ### Comment · Reviewer_KF6g · 2023-08-14
> >
> > Thank you for your response. I will be keeping my score as it is.

---

### Official Review · Reviewer_Gzxd · 2023-07-05

**Soundness:** 2 fair
**Presentation:** 3 good
**Contribution:** 2 fair
**Rating:** 6
**Confidence:** 3

**Summary:**

This paper studies a contextual Bayesian optimization problem when the true and reference distributions of the context can be different due to distribution shifts. The authors propose an algorithm called robust Bayesian satisficing algorithm (RoBOS) based on the idea of robust saisificing (RS). Through theoretical analysis and empirical results, the authors demonstrate their results on two notions of regret: lenient regret and robust satisficing regret. For the theoretical part, the authors have a thorough analysis and show that RoBOS achieves with high probability $\tilde{\mathcal{O}}(\gamma_T \sqrt{T})$ robust satisficing regret and  $\tilde{\mathcal{O}}(\gamma_T \sqrt{T} + \sum_{t=1}^T \epsilon_t)$ lenient regret, where $\gamma_T$ is the maximum information gain and $\epsilon_t$ is the amount of distribution shift in round $t$. For the empirical part, the authors propose two synthetic benchmarks and one real-world benchmark, and for all cases, they demonstrate RoBOS outperforms distributionally robust BO (DRBO).

**Strengths:**

Distribution shift is an important challenge in Bayesian Optimization, and it is great authors propose a new algorithm that attempts to address these issues, it also brings in some new interesting future research questions. The proposed question and results are new, to my best knowledge. I also like the algorithm, which naturally combines the empirical fragility and common algorithms in the contextual Bayesian optimization setting.  The paper is well-written in most parts except for a few minor parts.



**Weaknesses:**

My biggest concern is the two regrets defined in the paper: the lenient regret and the robust satisficing regret, which depends explicitly on the threshold $\tau$. In particular, previous papers in robust satisficing [1] also did the analysis using common performance measures such as average performance. The authors also lack justification for their proposed new notion: robust satisficing regret, beyond matching the definition of robust satisficing which might favor RoBOS. In addition, think the theoretical analysis employed is standard in the literature.

I think the following issues need to be addressed:
1. Could you also demonstrate (theoretically or empirically) the performance of RoBOS if the goal is to maximize the reward?
2. How sensitive is the choice of $\epsilon$ in DRBO/ WRBO versus the choice of $\tau$ in RoBOS?
In the literature, e.g., Figure 1 of [1], they measured the performance of the robust satisficing model over a sequence of $\tau$ and solve the distributional robust optimization model over a sequence of radius of $r$, then they compared the efficient frontier on both average performance and CVaR. This might be a way to further justify the effectiveness of RoBOS.

Reference:

[1] Daniel Zhuoyu Long, Melvyn Sim, and Minglong Zhou. Robust satisficing. Operations Research, 71(1):61–82, 2023.


**Questions:**

There are a few places where terms are not defined in the statement, or they are confusing:
1. In lines 102 and 103, can you be more specific on the difference between $\kappa_{\tau,t}$ and $\kappa_{\tau,t}(x)$? In particular, in both optimization problems, what are the decision variables, and what is fixed?
2. In Lemma 1 (line 155), the term $\gamma_{t-1}$ is not defined, the term appears in line 185, which should be moved earlier.
3. The definition of $B’$  should be moved from line 227 to the statement of Theorem 4.
4. In Figure 3, the first line, should be benchmark “2”.


Questions for the empirical results:
1. In Figure 1, why $\tau = Z_0/2$?
2. In line 256, the true distribution is picked randomly from the set $\mathcal{U}_t$, can you be more specific on this?
3. In Figure 3, the latter 2 plots seem incomplete. For lenient regret, why does the plot only plot for $t \le 100$? For robust satisficing regret, only RoBOS is complete.
4. In both synthetic benchmarks, could you justify the choice of $\tau$? In particular, are the experimental results consistent for a wide range of $\tau$?

Questions for the theoretical results:
1. Page 10 of the appendix, line 288: is the reward function $r(t)=-|o(t-K)_+|$?
2. How is Eq (29) in line 323 derived? Why can we set $B=1$ in Lemma 5? It might be good to be more specific here.



**Limitations:**

The authors stated their proposed method RoBOS may be worse than DRBO if the goal is to maximize the reward, which is the main limitation of their work. The other potential limitation I think is the robust satisficing model might be sensitive to the choice of $\tau$.

I suggest the author also analyze their results in other performance measures and see how sensitive their results are with respect to the choice of $\tau$.

There is no potential negative societal impact.

---

> ### Author Rebuttal · Authors · 2023-08-09
>
> We thank the reviewer for their thoughtful comments and valuable insights.
>
> ### Weaknesses
>
> - **(...justification for robust satisficing regret)**
> Robust satisficing regret evaluates how our algorithm fares against the robust satisficing action. In particular, the true robust satisficing action $x^*\_t$ achieves
> \begin{equation*}
>     \mathbb{E}\_{c\sim P^*\_t}[f(x^*\_t,c)] \geq \tau - \kappa\_{\tau,t} \Delta(P^*\_t,P\_t) ~.
> \end{equation*}
> Robust satisficing regret, evaluated at each round, measures the expected reward of the action chosen by our algorithm against the guaranteed reward of $x^*\_t$. When the selected action meets or exceeds this guaranteed reward, no regret is accumulated. However, if the chosen action falls short of this guarantee, then regret is accrued. Based on the above discussion, we think that robust satisficing regret is the right choice when evaluating the loss of the learner w.r.t. robust satisficing benchmark. That being said, we agree with the reviewer about the importance of investigating the cumulative reward performance of RoBOS. Even though our goal is to satisfice the aspiration goal, it is still meaningful to analyse the cumulative reward. For instance, Figure 1 of the main paper shows a scenario in which the robust satisficing solution attains the optimal reward at the true distribution $P^*\_t$, while DRO with imprecise uncertainty sets cannot do so. Since RoBOS converges to the robust satisficing solution, in this example RoBOS is expected to attain higher cumulative reward than DRBO.
>
> - **(...performance of RoBOS in reward maximization...)**
> During the response phase, we compared RoBOS and DRBO in terms of their cumulative rewards. The results can be found in the response pdf.
>
> - **(...sensitivity to $\tau$...)**
> During the response phase, we compared RoBOS and DRBO in terms of their sensitivity to the choice of $\tau$ and $\epsilon$. The results can be found in the response pdf.
>
>
> ### Questions
> #### Definitions
> 1) $\kappa\_{\tau,t}$ and $\kappa\_{\tau,t}(x)$ can be defined as following. In RS, the objective is to find $x^*\_t \in \mathcal{X}$ that solves in each round $t$
> \begin{equation*}
> \kappa\_{\tau,t} = \min k ~ \text{s.t.} ~ \mathbb{E}\_{c\sim P}[f(x,c)] \geq \tau - k \Delta(P,P\_t), ~ \forall P \in {\cal P}\_0 ~ , x \in {\cal X}, ~ k \geq 0 ~ .
> \end{equation*}
> To find $x^*\_t$, we can first compute the {\em fragility} of $x \in {\cal X}$ as
> \begin{equation*}
> \kappa\_{\tau,t}(x) = \min k ~ \text{s.t.} ~ \mathbb{E}\_{c\sim P}[f(x,c)] \geq \tau - k \Delta(P,P\_t), ~ \forall P \in {\cal P}\_0, ~ k \geq 0 ~ .
> \end{equation*}
> In the first problem $\tau$ and $P\_t$ are fixed while the decision variable is $x\in\mathcal{X}$. In the second problem, $x$, $\tau$ and $P\_t$ are fixed and we compute the minimum $k$ that satisfies the constraints.
> 2) Fixed.
> 3) Fixed.
> 4) Fixed.
>
> #### Empirical
> 1) Figure 1 illustrates a proof of concept example where $\tau$ is chosen to be achievable for only one action, highlighted by the green rhombus.
> 2) In the second synthetic experiment we pick the true distribution $P^*\_t \sim \mathcal{N}(\mu\_t +  N\_t, 5^2)$ where $\mu\_t$ is the mean of the reference distribution $P\_t$ and $N\_t\sim \mathcal{U}(-6,6)$ is set as to  make sure the distributional shift is not greater than $\epsilon$.
> 3) We appreciate your careful review of our work. However, we'd like to clarify that all our plots do extend until $t=200$. It's possible that there may have been some technical issues with the visualization during the pdf loading process. We recommend viewing the file in a different pdf reader. We apologize for any inconvenience caused and we are here to assist if further clarifications are needed.
> 4) In the first proof of concept synthetic experiment, $\tau$ is chosen to be achievable only by one of the actions. In the second synthetic experiment, $\tau$ was selected as half of the function maximum arbitrarily. Further, a discussion about how to select $\tau$ in a real-world setting is given in our global response. We have made revisions in response to your comments and have conducted an additional experiment to analyse RoBOS' sensitivity to $\tau$. We also compare this with DRBO's sensitivity to the choice of $\epsilon$. These results are included within the response pdf.
>
> #### Theoretical
> 1) The pseudo reward function we defined for the insulin dosage allocation experiment is $-|o(t) - K|$. Here $o(t)$ is the blood glucose level of the patient, measured 150 minutes after their meal, and  $K=112.5$ mg/dl is the target blood glucose level. Our pseudo reward function measures the discrepency in patients blood glucose compared to the target level.
>
> 2) In the mentioned equation $\langle w^*\_t, \sigma\_t(x\_t,\cdot)^2\rangle$ corresponds to $m\_t$ in Lemma 5. Hence $S\_t$ correspods to $\sigma\_t(x\_t, c\_t)^2$, $c\_t\sim P^*\_t$. Due to our assumption on the kernel that $\sigma(x\_t,c\_t) \leq 1$, $B$ in Lemma 5 can be set to 1.

---

> > ### Comment · Reviewer_Gzxd · 2023-08-14
> >
> > I thank the authors for their detailed response that addresses my concerns.
> > I changed my rating to 6.

---

### Official Review · Reviewer_bA8n · 2023-07-06

**Soundness:** 3 good
**Presentation:** 2 fair
**Contribution:** 2 fair
**Rating:** 5
**Confidence:** 4

**Summary:**

This paper studies robust satisficing in contextual Bayesian optimization under distribution shift in the distributions of the context.
They show that under some assumptions their algorithm achieves sublinear lenient regret and under some relaxed assumptions they achieve sublinear robust satisficing regret.

They compare their method agains distributionally robust optimization approach [10] for contextual Bayesian optimization.

**Strengths:**

I have my reservations against points that I mentioned below, but I believe this research direction is valuable and interesting.

**Weaknesses:**

The writing of the paper can be improved. The clarity and coherence of the writing could be enhanced. There are instances where the ideas are not effectively communicated, leading to confusion for the reader. Furthermore, the structure of the paper could benefit from more logical organization and smoother transitions between sections. Additionally, some grammatical and punctuation errors need to be addressed, as they can detract from the overall quality of the paper. A revision focusing on refining the writing style and ensuring a more polished presentation would greatly strengthen the paper.

**Questions:**

1) Why did you choose to use MMD distance to measure the discrepancy between distributions.
2) In Table 1, there must be a space between $\min$ and $k$ for RS objective.
3) I do not fully understand why one would prefer RS over DRO. Can you please clarify this point? When it would make sense to use RO, RS and DRO?
4) Line 141: Please provide references that GP assumption is common.
5) Lemma 1: Where does the det operator appear appear in Lemma 1? Why is it introduced there? What is $\gamma_{t-1}$? What is $e$?
6) If LCB is never used, why it is introduced?
7) How restrictive the regularity assumption on $f$?
8) Optimism in the face of uncertainty is a well-known concept but I am not sure how it is motivated in Section 3 and how your model is optimistic. Can you please elaborate on this?
9) For the fair comparison of methods: To make a fair comparison between DRBO and your method you should consider robust regret introduced in [10].
10) Is $\tau$ fixed throughout the algorithm? Referring to footnote at Page 7, I understand that you claim $\tau$ can be selected adaptively but I think it must be selected adaptively otherwise $\hat \kappa_{\tau, t}(x)$ can be infinity, right?
11) Can you please intuitively explain what does the model in (1) satisfy and what is it robust against? Why $\tau$ can be explained as $Z_t?
12) I find it unnatural that the regret bound in Theorem 3, is not dependent on $\tau$. I understand it is because $\tau$ appears also in the RS regret definition. I think a more natural approach would be separating these two $\tau$'s and seeing that the regret only depends on the difference of these two $\tau$'s. (similar argument goes for the lenient regret) Can you also show the regret guarantee on the robust regret as defined in [10]?

**Limitations:**

Yes.

---

> ### Author Rebuttal · Authors · 2023-08-09
>
> We thank the reviewer for their thoughtful comments and valuable insights.
>
> ### Weaknesses
> **(...refining the writing style and ensuring a more polished presentation...)**
> We have undertaken a thorough revision to address the concerns raised. Specifically, we've refined the writing for better clarity, restructured the content for a more logical flow, and ensured smoother transitions between sections.
>
> ### Questions
> 1) MMD distance, defined using a kernel function, is a prevalent measure for distributional shifts in RKHS [A, B, 10]. Its overlap with kernel-based methods is notable, and with kernels like RBF and Matérn-$\nu$, MMD acts as a metric. Its ease of computation and clear interpretation aids theoretical analysis.
> 2) Fixed.
> 3) We kindly refer the reviewer to our global response.
> 4) Some of the work that utilizes GP's in the RKHS setting are [1, 10, 31] which are also referred to in the main paper. Now we also cite them in the GP assumptions section.
> 5) There was a writing error, now fixed and confidence bounds updated to a more recent version. In the final version $\gamma\_t$ (the maximum information gain) is defined in its proper place.
> 6) Fixed.
> 7) The assumption that $f$ belongs to an RKHS with bounded Hilbert norm is a very common one in the literature [1, 31]. The RKHS assumption induces smoothness conditions
> \begin{equation*}
>     |f(x) - f(y)| = |\langle f, k(x,\cdot) - k(y,\cdot) \rangle| \leq \lVert f\rVert\_{\mathcal{H}} \lVert k(x,\cdot) - k(y,\cdot)\rVert
> \end{equation*}
> by Cauchy-Schwarz inequality.
> 8) RoBOS's optimism arises from its use of the UCB of the objective function, similar to algorithms like GP-UCB, ensuring a balance between exploration and exploitation. Lemma 2 emphasizes this: with a probability of at least $1-\delta$, we have $\hat{\kappa}\_{\tau,t}(x) \leq \kappa\_{\tau,t}(x)$ for all $x\in\cal X$ and $t \geq 1$. This indicates that the algorithm's estimated fragility is always less than or equal to the actual fragility, confirming RoBOS's optimistic nature.
> 9) In the response period, we conducted experiments comparing DRBO and RoBOS on the robust regret from [10], figures given in the response pdf. Our findings show that RoBOS, even with linear robust regret, can outperform DRBO in cumulative reward. This occurs when the ambiguity set isn't a tight representation of the distributional shift, leading the DRO solution DRBO converges to, to be suboptimal under the true distribution.
> 10) $\tau$ can indeed be fixed or time-varying, note that our algorithm works even when the $\hat{\kappa}\_{\tau,t}(x)$ is infinite for some $x\in \mathcal{X}$. We only require that the robust satisficing problem is feasible, meaning the following optimization objective has a solution. Find $x^*\_t \in \mathcal{X}$ that solves in each round $t$
> \begin{equation}
>     \kappa\_{\tau,t} = \min k ~ \text{s.t.} ~ \mathbb{E}\_{c\sim P}[f(x,c)] \geq \tau - k \Delta(P,P\_t), ~\forall P \in {\cal P}\_0 ~, x \in {\cal X}, ~k \geq 0 ~. \tag{1}
> \end{equation}
> which is feasible under $\tau \leq \mathbb{E}\_{c\sim P\_t}[f\_{\hat{x}\_t}]$, for all $t\in [T]$, where $\hat{x}\_t := \arg\max\_{x \in {\cal X}} \langle w\_t, f\_x \rangle$. If this assumption does not hold in round $t$, then the robust satisficing problem is infeasible, which means that $\kappa\_{\tau,t} = \infty$ and there is no robust satisficing solution. Therefore, measuring the regret in such a round will be meaningless. In practice, if the learner is flexible about its aspiration level, this assumption can be relaxed by dynamically selecting $\tau$ at each round to be less than $\langle w\_t, \text{lcb}^t\_{\hat{x}'\_t} \rangle$, where $\hat{x}'\_t := \arg\max\_{x \in {\cal X}} \langle w\_t, \text{lcb}^t\_x \rangle$. The optimism principle ensures that if the primary problem (1) is feasible, the problem with estimated fragility $\hat{\kappa}{\tau,t}$ remains so. Given feasibility, an expert can set the satisficing goal $\tau$, see our example on diabetes.
> 11) The model in (1) tries to find a solution that achieves the desired threshold $\tau$, i.e.\ satisfice. The model picks the solution that has the lowest fragility $\kappa\_{\tau,t}(x)$. The fragility can be viewed as the minimum rate of suboptimality one can obtain with respect to the threshold, per unit of distribution shift from $P\_t$. Since the fragility is calculated over all possible context distributions $\forall P \in {\cal P}\_0$, the model gives robustness guarantees under all possible distributional shifts. In particular, the true robust satisficing action $x^*\_t$ achieves
> \begin{align*}
>     \mathbb{E}\_{c\sim P^*\_t}[f(x^*\_t,c)] \geq \tau - \kappa\_{\tau,t} \Delta(P^*\_t,P\_t) ~
> \end{align*}
> under the true distribution $P^*\_t$, no matter what the distributional shift is. $\tau$ can be explained as a percentage of $Z\_t$ when $f$ is known (not in the learning problem, where $f$ is unknown). Only knowing $Z\_t$ (but not $f$ or optimal $x$), the decision-maker can set $\tau$ as the percentage of $Z\_t$ that it is content with. This is just one example of how $\tau$ can be selected. See our response to your previous comment for another example.
> 12) The $\tau$ value found within our regret definitions is identical to the $\tau$ utilized as an input to our algorithm. It represents the aspiration level that our algorithm strives to reach, and forms the benchmark against which we define our regret notions. Its absence in the regret bound is due to its direct link with the objective function $f$, assuming $\tau \leq \mathbb{E}\_{c\sim P\_t}[f\_{\hat{x}\_t}]$.

---

> > ### Comment · Reviewer_bA8n · 2023-08-14
> > **Response to the author rebuttal**
> >
> > I thank the authors for their detailed response. I have some further comments that I realize when I was going over the paper.
> > * In Figure 1, is there a typo in the definition of $Z_0 = \mathbb E_{c \sim P_t}$?
> > * I am still having hard time understanding why $\tau$ can be explained as a percentage of $Z_t$ when $f$ is known.
> > * I do not see that the assumption $\tau\leq \mathbb E_{c \sim \mathbb P_t}[f_{\hat x_t}]$ in the statement of Theorem 3. In Line 190, it is stated that Theorem 3 holds for any threshold $\tau$. Can you please clarify these points?
> >
> > Thank you!

---

> > > ### Author Response · Authors · 2023-08-14
> > >
> > > We thank the reviewer for their careful reading of our paper and the new comments.
> > >
> > > 1. Yes, $Z\_0$ should be $Z\_t = \max\_{x \in {\cal X}} \mathbb{E}\_{c \sim P\_t} [f(x,c)]$. We will fix this typo.
> > >
> > > 2. The robust satisficing optimization problem (given after line 102) at round $t$ is feasible when $\tau <= Z\_t$. When $\tau > Z\_t$, the problem is infeasible, and there is no robust satisficing action $x^*\_t$. Since the learner receives the reference distribution $P\_t$ at the beginning of each round, when $f$ is known, $Z\_t$ can be computed exactly by the learner. Since the true distribution $P^*\_t$ and the amount of distribution shift is unknown, the optimal action (with expected reward $Z\_t$) under $P\_t$ can be far from optimal under $P^*\_t$. To protect against distribution shifts, the learner can solve the robust satisficing problem. For instance, if the learner is content with receiving 90% of $Z\_t$ under no distribution shift, it can set $\tau = 0.9 Z\_t$ and solve for $x^*\_t$. Now, $x^*\_t$ will offer an expected reward that is at least $0.9 Z\_t$ if $P^*\_t = P\_t$. If $P^*\_t \neq P\_t$, $x^*\_t$ will offer an expected reward that is at least $0.9 Z\_t - \kappa\_{\tau,t} \Delta(P^*\_t, P\_t)$. As an example of why the learner can be content with 90% of $Z\_t$, see our diabetes example in the general response and classifier example given in response to Reviewer gjzH.
> > > One may also wonder what will change if the learner sets $\tau' = 0.95 Z\_t$. In this case, the fragility under $\tau'$, i.e., $\kappa\_{\tau',t}$ can be higher than $\kappa\_{\tau,t}$, which will result in diminished expected reward guarantees under large distribution shifts.
> > >
> > > 3. Thanks for noticing this. As we mentioned in our response above, when $\tau > \mathbb{E}\_{c \sim P\_t} [f\_{\hat{x}\_t}] = Z\_t$, the robust satisficing problem is not feasible, i.e., $\kappa\_{\tau,t} = \infty$. If this is the case, by looking at (3), one can say that by convention, the regret in round $t$ is $0$ independent of the chosen action $x\_t$. Since there is no robust satisficing action $x^*\_t$ in round $t$, there is no way we can evaluate the loss of the learner with respect to the robust satisficing action in round $t$.
> > > So, by definition, when $\tau > Z\_t$, the regret will be $0$ and the regret bounds will still hold.
> > > To improve the clarity of the paper, we will explain this in the statement of Theorem 3 in the revised paper.
> > >
> > > We hope that our response above has clarified your concerns. If you have any other comments, we will happily address them.

---

> > > > ### Comment · Reviewer_bA8n · 2023-08-20
> > > >
> > > > Thank you very much for your response!
> > > > I suggest to include the additional experiments also in the revised paper.
> > > >
> > > > I will increase my score to 5.

---

> > > > > ### Author Response · Authors · 2023-08-20
> > > > >
> > > > > Thank you very much for your valuable feedback. We will include the additional experiments in the revised paper.

---

### Official Review · Reviewer_gjzH · 2023-07-07

**Soundness:** 3 good
**Presentation:** 3 good
**Contribution:** 3 good
**Rating:** 6
**Confidence:** 3

**Summary:**

I think the main contributions of this paper are as follows:

- Proposes a new decision-making framework called robust Bayesian satisficing (RBS) which combines robust satisficing with Bayesian optimization. RBS aims to achieve a satisfactory solution under distributional shifts by observing a predefined satisfactory threshold. This is different from distributionally robust optimization which requires an ambiguity set and stochastic optimization which optimizes for a given reference distribution.

- Defines two regret measures to evaluate the performance of RBS algorithms: lenient regret and robust satisficing regret. Lenient regret measures the cumulative loss of an algorithm's chosen actions with respect to an aspiration level. Robust satisficing regret measures the loss with respect to the robust satisficing benchmark which is the aspiration level minus the fragility (a measure of suboptimality per unit distribution shift). The paper shows the connection between these two regret measures.

- Proposes an RBS algorithm called Robust Bayesian Optimistic Satisficing (RoBOS) which uses Gaussian processes to model the objective function. RoBOS only requires an aspiration level as input and does not need an ambiguity set. RoBOS chooses actions to minimize the estimated fragility which is an optimistic estimate of the true fragility.

- Proves that RoBOS achieves sublinear robust satisficing regret and lenient regret under certain assumptions. The regret bounds depend on the maximum information gain and the sum of distribution shifts.

- Demonstrates the effectiveness of RoBOS on synthetic problems and compares it with other robust Bayesian optimization algorithms.

**Strengths:**

Robust Bayesian satisficing is a novel framework that combines robustness to distributional shifts with satisficing behavior. This provides an alternative to existing paradigms like distributionally robust optimization and stochastic optimization. RBS does not require precise knowledge of the ambiguity set and can handle unknown distribution shifts. Also, the paper provides theoretical guarantees on the regret of RoBOS under some assumptions. The robust satisficing regret and lenient regret of RoBOS grow sublinearly with time. The regret bounds show the dependence on maximum information gain and the sum of distribution shifts, providing insight into how RoBOS handles distributional shifts. In addition, the experimental results verify the theoretical findings on the sublinearity of the regret bounds. Finally, the paper is well-written, clear, and easy to follow.

**Weaknesses:**

- Some assumptions seem to be strong for this problem. For example, the bounds require the sum of distribution shifts to be sublinear in time which may not always hold in practice. It would be good to discuss how the algorithm behaves when these assumptions are violated.

- The experiments are limited to synthetic problems. It would be good to evaluate RoBOS on some real-world benchmark problems to demonstrate its effectiveness in practical settings. Comparisons with more algorithms on these problems would also strengthen the experimental evaluation. There are many datasets for distributional shifts that the author can leverage to verify their algorithm.

- Although as defined in the conclusion as a future direction, it would be beneficial to discuss the effects of continuous context in this problem, since the focus is on deterministic contexts in this paper. Analyzing RoBOS when contexts are stochastically generated would provide greater insight into how it handles uncertainty.

**Questions:**

In addition to the previous section:
I am curious how this approach can be utilized for test-time distribution shifts and adaptations.

**Limitations:**

They covered this part very well in my view.

---

> ### Author Rebuttal · Authors · 2023-08-09
>
> We thank the reviewer for their thoughtful comments and valuable insights.
>
> ### Weaknesses
> - **(Some assumptions...)**
> Indeed to achieve sublinear lenient regret, some assumptions have to be made on the distribution shift. Nothing much can be said in an adversarial environment where the reference distribution $P\_t$ is judiciously chosen away from the true distribution $P^*\_t$ in each round. However, one interesting practical setting is the case where the reference distribution is the empirical distribution. This corresponds to data-driven optimization setting discussed in [10]. For this case, $P^*\_t = P^*$ is fixed for $t \in [T]$, $P\_1$ is the uniform distribution over the contexts, and $P\_t = \sum\_{s=1}^{t-1} \delta\_{c\_s}$, $t>1$ is the empirical distribution of the observed contexts, where $\delta\_x$ is the Dirac measure defined for $c \in {\cal C}$ such that $\delta\_c(A) = 1$ if $c \in A$ and $0$ otherwise. Under this setting, one can put a probabilistic bound on the amount of distributional shift $\epsilon\_t$, which can be used to bound the regret similarly to the proof of  [10, Corollary 4]. We are able to show that the lenient regret of RoBOS when run with $\beta (\delta/3)$ is bounded with probability at least $1-\delta$ by
> \begin{equation*}
>     R^{\textit{l}}\_T \leq 4 \beta\_T \sqrt{T\left( 2 \gamma\_T + 2\log \left(\frac{12}{\delta}\right)\right)} + B' \epsilon\_1 + B'  2 \sqrt{T} \left(2 + \sqrt{2 \log \left( \frac{\pi^2 T^2}{2\delta} \right)} \right) ~.
> \end{equation*}
>
> - **(The experiments are limited...)** We note that we included a real-world experiment in our supplemental document, now moved to the main paper, concerning insulin dosage allocation for Type 1 diabetes patients. For more detail about the experiment, we kindly refer the reviewer to our response to reviewer KF6g's first question.
>
> - **(...discuss the effects of continuous context...)**
> In the original paper, we assume that at each round $t\in[T]$, contexts are drawn stochastically from the true distribution $P^*\_t$ over a finite context set. We studied the continuous context case in the response phase, and observed that our regret bounds hold with only minor differences in the regret analysis. The proof for the continuous case closely follows the discrete case, however, instead of expressing the expectations as an inner product (e.g., $\langle w\_t, f\_x \rangle$), we work with the general representation (e.g., $\mathbb{E}\_{c\sim P\_t}[f\_x]$). Due to linearity of expectations, our derivations continue to hold. Also in the analysis of the lenient regret, in equation (37), instead of using the Cauchy-Schwarz inequality, we use the definition of the MMD distance to bound the term $\mathbb{E}\_{c \sim P\_t}[f(\hat{x}\_t, c)]  -  \mathbb{E}\_{c \sim \bar{P}^t\_{\hat{x}\_t}} [f(\hat{x}\_t,c)] \leq B \Delta(P\_t, \bar{P}^t\_{\hat{x}\_t})$, where $B$ is an upper bound on the RKHS norm of $f$ and $\Delta(\cdot,\cdot)$ is the MMD distance.
>
>
> ### Questions
> **(...test-time distribution shifts...)**
> Indeed a promising area of application is the test-time distributional shifts. RoBOS presents itself as a possible solution to this problem especially when the test-time distributional shift is difficult to foresee. Many engineering products are designed to be deployed in unpredictable environments with multiple contingent factors. The safety guarantees of RoBOS with respect to all possible distributional shifts may give it an edge in such environments. For instance, when $P\_t$ is the training distribution, one can train a model to achieve the maximum accuracy, i.e., $\hat{x}\_t = \arg\max\_{x \in {\cal X}} \mathbb{E}\_{c \sim P\_t}[f(x,c)]$. However, this will be of no use when the test distribution $P^*\_t$ is significantly different from $P\_t$. Instead, with robust satisficing, one seeks to achieve an accuracy that we are content with, e.g., $\tau=0.9$ (represents $90\\%$ accuracy). Then, the robust satisficing solution guarantees at test time, an accuracy that is at least $\tau - \kappa\_{\tau,t} \Delta(P^*\_t, P\_t)$.

---

> > ### Comment · Reviewer_gjzH · 2023-08-19
> >
> > Thanks to the authors for providing a detailed response. I have read the responses and other reviews, and still keep my score.

---

### Author Rebuttal · Authors · 2023-08-09

We sincerely thank the reviewers for their careful reading of our paper and constructive comments. Here we address the common questions raised by the reviewers.
- **(Real-world experiments and how to select $\tau$)** One real-world experiment concerning safe dose allocation for diabetes patients is presented in the supplementary document. Type 1 Diabetes Mellitus (T1DM) patients require bolus insulin doses (id) after meals for postprandial blood glucose (pbg) regulation. One of the most important factors that affect pbg is meal carbohydrate (cho) intake [C]. Let ${\cal X}$ and ${\cal C}$ represent admissible id and cho values. For $x \in {\cal X}$, $c \in {\cal C}$, let $g(x,c)$ represent the corresponding (expected) bpg value. Function $g$ depends on the patient's characteristics and can be regarded as unknown. The main goal of pbg regulation is to keep pbg close to a target level $K$ in order to prevent two potentially life-threatening events called hypoglycemia (e.g., pbg $<$ 70 mg/dl) and hyperglycemia (e.g., pbg $>$ 180 mg/dl). This requires $x\_t$ to be chosen judiciously based on current $c\_t$. Patients rely on a method called cho counting to calculate $c\_t$. Often times, this method is prone errors [D]. The reported cho intake $\zeta\_t$ can differ significantly from $c\_t$. In order to use DRO, one needs to identify a range of plausible distributions for cho calculation errors, which is hard calculate and interpret. On the other hand, specifying $\tau$ corresponds to defining an interval of safe pbg values around $K$ (e.g., pbg $=$ 125 mg/dl) that one is content with, which is in line with the standard clinical practice [E]. We will move this experiment from the supplemental document to the main paper. In addition, in response to the comments of reviewer Gzxd and KF6g, we performed new experiments that compare accumulated rewards of RoBOS and other competing benchmarks. Moreover, we also carried out simulations that shows sensitivity of the results to the aspiration level $\tau$ set by the learner. These results can be found in the response pdf.
- **(Advantages of RS over DRO)** RS has multiple advantages over DRO depending on the problem setup. The main difference is that it considers the goal of satisficing rather than optimizing. Satisficing is argued to be normatively better than optimizing by many decision theorists and philosophers [16, 17, 18], especially under settings of deep uncertainty. Once the normative shift is made from optimizing to satisficing, a reasonable approach for decision-making is to maximize the robustness of the action to the uncertainties of the environment. RS achieves this by selecting the action that achieves the desired threshold under all possible context distributions the environment can produce. When no such action is present, RS picks the action that achieves the threshold under the biggest set of context distributions and minimizes the margin of error with respect to the threshold. Consider the case where $\tau = Z\_r := \max\_{x \in \mathcal{X}}\min\_{P\in\mathcal{U}(r)} \mathbb{E}\_{c\sim P} [f(x,c)] $, where $\mathcal{U}(r)$ is the ambiguity ball of radius $r$ centered at $P\_t$. For this setting DRO solution implies
\begin{align*}
    &\tau - \mathbb{E}\_{c\sim P} [f(x\_{\text{DRO}},c)] \leq 0 \qquad \forall P \in \mathcal{U}(r) \\\\
    &\tau - \mathbb{E}\_{c\sim P} [f(x\_{\text{DRO}},c)] \leq \infty \qquad \forall P \in {\cal P}\_0 \setminus \mathcal{U}(r)
\end{align*}
whereas the RS solution implies
\begin{align*}
    &\tau - \mathbb{E}\_{c\sim P} [f(x\_{\text{RS}},c)] \leq \kappa\_\tau \Delta(P,P\_t) \qquad \forall P \in \mathcal{U}(r) \\\\
    &\tau - \mathbb{E}\_{c\sim P} [f(x\_{\text{RS}},c)] \leq \kappa\_\tau \Delta(P,P\_t) \qquad \forall P \in {\cal P}\_0 \setminus \mathcal{U}(r)
\end{align*}
Algorithmically, RoBOS needs less information than DRBO since unlike DRBO, RoBOS does not utilize an ambiguity set for which the true distribution is assumed to be in. It is shown in the experiments that when the ambiguity ball is not selected correctly, DRBO solutions can be suboptimal or even disastrous w.r.t. the satisficing goals. In contrast, it would be better to use DRO if we have a tight and trustable ambiguity set and the goal is optimizing rather than satisficing. In addition, in some applications such as blood glucose regulation, RS is more interpretable than DRO (see our previous response).

### References
- [A] Gretton et al., A kernel two-sample test. 2012.
- [B] Sejdinovic et al., Equivalence of distance-based and rkhs-based statistics in hypothesis testing. 2013.
- [C] Walsh et al., Guidelines for optimal bolus calculator settings in adults. 2011
- [D] Kawamura et al., The factors affecting on estimation of carbohydrate content of meals in carbohydrate counting. 2015.
- [E] Kahanowitz et al., Type 1 diabetes–a clinical perspective. 2017.

---

### Decision · Program_Chairs · 2023-09-21

**Decision:**

Accept (poster)

**Comment:**

This work studies (contextual) Bayesian optimization with shifts in the distribution of contexts. The authors propose a new algorithm motivated by the idea of robust satisificing. The paper makes meaningful algorithmic and analytic contributions to multiple communities (Bayesian optimization, distribution shifts). The reviewing team has raised several issues with the exposition. In addition to addressing the carefully, I recommend the authors to candidly discuss the limitations of their work in the camera-ready version.